**Analysis**

# Comprehensive benchmarking and guidelines of mosaic variant calling strategies

Yoo-Jin Ha[1,2], Seungseok Kang [1,2], Jisoo Kim [1], Junhan Kim[1], Se-Young Jo [1,2] & Sangwoo Kim [1,2,3] ✉

Rapid advances in sequencing and analysis technologies have enabled the accurate detection of diverse forms of genomic variants represented as heterozygous, homozygous and mosaic mutations. However, the best practices for mosaic variant calling remain disorganized owing to the technical and conceptual difficulties faced in evaluation. Here we present our benchmark of 11 feasible mosaic variant detection approaches based on a systematically designed whole-exome-level reference standard that mimics mosaic samples, supported by 354,258 control positive mosaic single-nucleotide variants and insertion-deletion mutations and 33,111,725 control negatives. We identified not only the best practice for mosaic variant detection but also the condition-dependent strengths and weaknesses of the current methods. Furthermore, feature-level evaluation and their combinatorial usage across multiple algorithms direct the way for immediate to prolonged improvements in mosaic variant detection. Our results will guide researchers in selecting suitable calling algorithms and suggest future strategies for developers.

Postzygotic mutations continuously occur along the zygote-to-adult trajectory, resulting in genetic mosaicism. Recently, the capabilities to examine the mosaic mutations at the genome level have led to a series of discoveries, including the mutational processes and landscapes involved in the development[1–3] and aging[4] of human and nonhuman organisms[5], the causes of neurological disorders[6] and cancer predispositions[7]. As research questions are being answered, growing attention and demand are required for the complete investigation of mosaicism, warranted by the accurate detection of mosaic mutations.

The detection of mosaic mutations is an intricate process, from its conceptualization. Given the definition of genetic mosaicism (that is, the presence of two or more genotypes), the scope of mosaic mutations has been loosely defined; conventional somatic mutations can also be seen as mosaicism because they cause genetic differences within an

individual. Moreover, severe difficulties lie in the technical side that confound the methodological principles. For example, mosaic mutations acquired during development may be present in multiple tissues, complicating the use of controls. Variant allele frequencies (VAFs) of mosaic mutations appear in a broad range, from extremely low (less than 1%) to the level of heterozygous germline variants (approximately 50%), depending on the time and location of the occurrence and can be also largely unbalanced among shared tissues[2]. This ambiguity is well reflected in the usage of a disparate set of approaches applied for recent studies, such as targeting variants with unlikely VAFs for normal zygosity in a single sample[8,9], searching for shared variants in a pair of samples[10] and taking advantage of machine-learning algorithms[11,12]. These circumstances urgently demand rigorous cataloging and assessment of mosaic detection algorithms, as has been done for germline and somatic variants[13–18], but it requires a more sophisticated design

[1]Translational Genome Informatics Laboratory, Department of Biomedical Systems Informatics, Yonsei University College of Medicine, Seoul, Republic of Korea. [2]Brain Korea 21 PLUS Project for Medical Science, Yonsei University College of Medicine, Seoul, Republic of Korea. [3]POSTECH Biotechnology Center, Pohang University of Science and Technology, Pohang, Republic of Korea. ✉e-mail: swkim@yuhs.ac

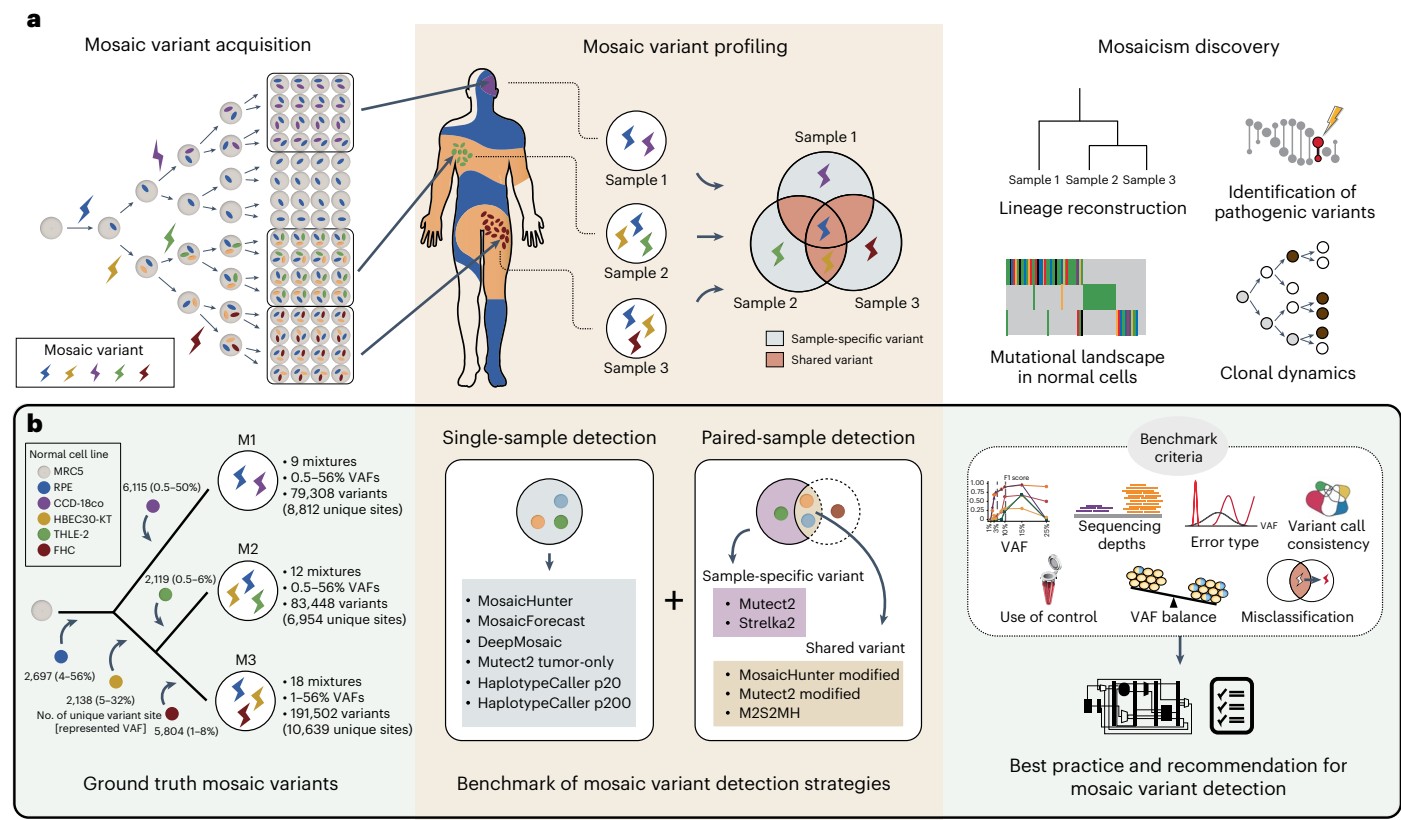

**Fig. 1 | A benchmark overview. a**, Mosaic variant acquisition from a zygote and its representation in an individual is shown. Mosaic variants can exist in sample-specific or shared form. **b**, Cumulative mixing of six pregenotyped normal cell lines to construct the ground truth, mimicking the mosaic variant acquisition process during early development. Thirty-nine mixtures are categorized into three mosaic types, M1–M3, depending on the compositions of the mixed cell lines. Eleven detection strategies were evaluated in single sample and/or paired sample under various criteria such as VAF, sequencing depth, error type, call set consistency, use of control, VAF balance and misclassification. Images of the human body outline, blood in a tube and DNA were adapted from Adobe Stock (stock.adobe.com) under an extended license.

and a reliable ground truth to cover the full extent of scenarios that mosaic variants can represent.

Here, we present a robust benchmark of 11 state-of-the-art mosaic single-nucleotide variant (SNV) and insertion-deletion (INDEL) detection approaches based on a comprehensive reference standard that serves a ground truth. The benchmark and the reference standard have been systematically designed to evaluate the performance of the algorithms concerning multiple conditions that users will face in their analyses; these include VAFs, sequencing depth, variant types, variant sharing, VAF balances and the use of matched controls. We provide measured accuracies, strengths and weaknesses of the algorithms, as well as the best detection practice in different conditions. Finally, we suggest potential strategies for improved detection of mosaic mutations that can be applied at the user and developer levels.

## Results

### Benchmark setting

The benchmark dataset was prepared using a recently constructed reference standard[19]. Briefly, the reference standard material is a collection of 39 mixtures of six pregenotyped normal cell lines. When mixed, germline SNVs and INDELs of the cell lines form mosaic-like mutations of a wide VAF spectrum (0.5–56%) that can be used as the ground truth. Likewise, confirmed nonvariant sites (reference homozygous) can be seen as control negatives. We designed the cell line mixing procedure to mimic the mutation acquisition process under cellular differentiation, such as early embryonic development (Fig. 1a,b, left).

The 39 mixtures are categorized into three types, M1 (nine mixtures), M2 (12 mixtures) and M3 (18 mixtures) based on the cell line combinations, each of which has different sets of mutations (8,812, 6,954 and 10,639 mutations in M1, M2 and M3). Biologically, these categories symbolize three distinct descendants, which contain both common and lineage-specific mosaic variants of diverse VAFs acquired from cumulative cell line mixing. Every mixture in a same category shares the mutation sites, but the final mutations are represented in different VAFs depending on the cell line compositions. Consequently, we secured 345,552 and 8,706 high-confidence SNVs and INDELs for control positives, and 33,093,574 nonvariants and 18,151 germline sites for control negatives from the 39 mixtures (Methods); the large number of control positives required a postrecalibration process for precision and F1 score, which we applied in the evaluation (Methods). The control positive variants were enriched in low VAF (70% of the variants under 10% VAF) to reflect the currently known VAF distribution of mosaic variants (Extended Data Fig. 1 and Supplementary Table 1).

Comparison between the mixtures of different categories enables testing for both shared and sample-specific mosaic variants of different VAFs. Selecting a pair of mixtures, out of 741 possible combinations (Methods), covers nearly complete scenarios for mosaic mutation detection, regarding variant sharing and its compositions. Sequencing data from deep whole-exome sequencing (1,100×) and its multistep-down sampling (125×, 250× and 500×) of the 39 mixtures were used for evaluation.

We selected 11 mosaic detection strategies for evaluation. The inclusion criteria were: (1) algorithms that explicitly aim to detect

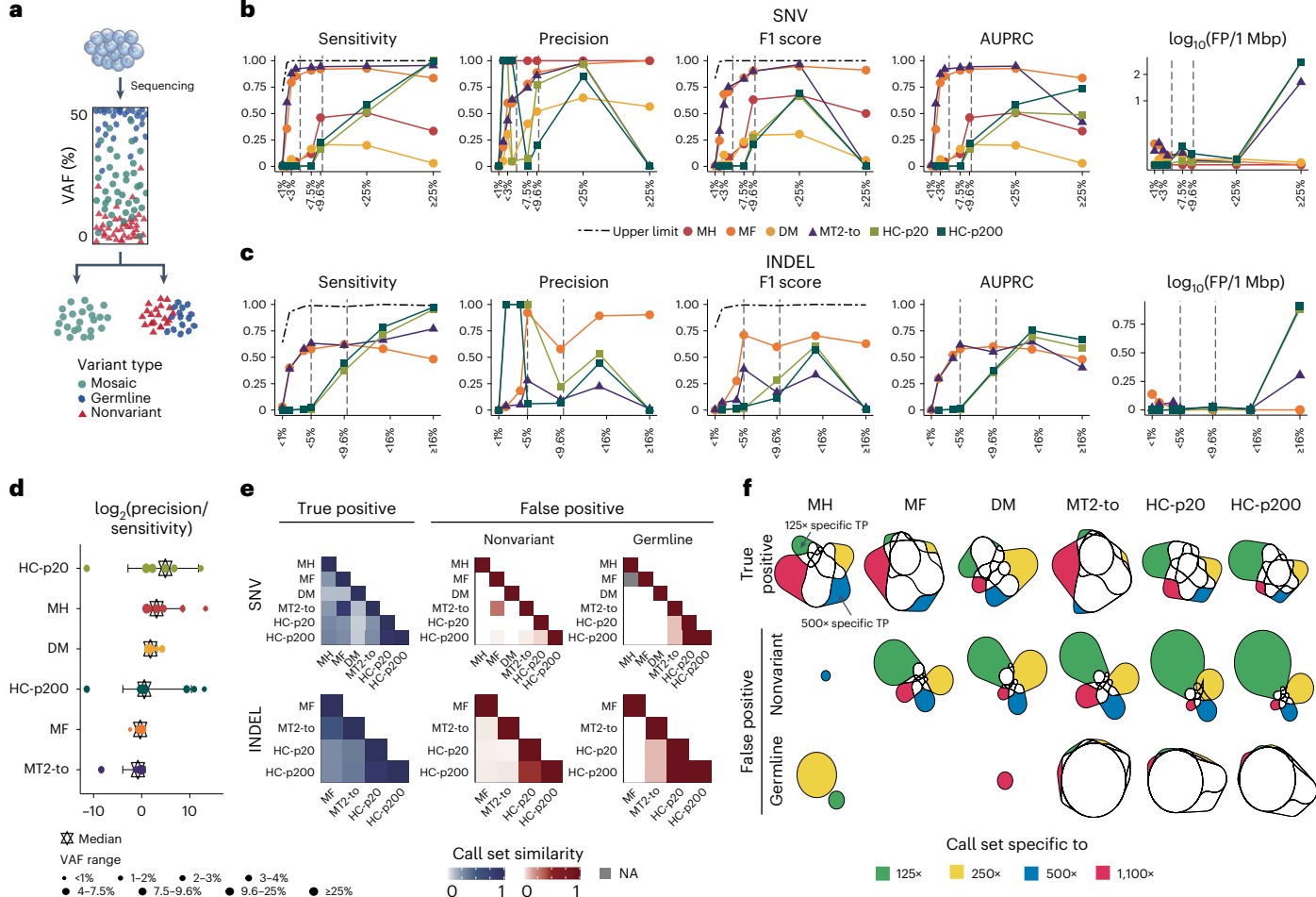

**Fig. 2 | Evaluation of single-sample-based calling. a**, A schematic overview of mosaic variant detection in a single sample. True mosaic variants with VAFs should be distinguished from the germline variants and diverse nonvariant sites. Thirty-nine samples of truth sets were used for evaluation. **b**,**c**, Sensitivity, precision, F1 score, AUPRC and the false positive rate ($\log_{10}$FP/1Mbp) in different VAF bins categories are shown. The dashed line with dots depicts the theoretical upper limit for sensitivity and F1 score by depletion of mutant alleles in sequencing data at low VAFs (Supplementary Notes). The two vertical lines with gray dashes refer to VAFs of 5 and 10%. The $y$ axis of $\log_{10}$(FP rate) is shown as a square root. **b**, SNV detection performance of six applied approaches in the eight VAF bins (less than 1, 1–2, 2–3, 3–4, 4–7.5, 7.5–9.6, 9.6–25 and ≥25%) with 1,100× data. **c**, INDEL detection performance of four applied approaches in the seven VAF bins (less than 1, 1–2, 2–4, 4–5, 5–9.6, 9.6–16 and ≥16%) with 1,100× data. **d**, The $\log_2$ ratios of precision

to the sensitivity are shown by using 39 truth sets of the six SNV callers. The bars depict the first and third quantiles and hexagonal stars show medians. **e**, Similarity of call sets between different approaches. The similarities were calculated using the Jaccard index. NA shows when the union of the call sets did not apply (or exist). **f**, Euler diagram illustrating the relative size and the relationships between sets shows the inconsistency of the variant call sets within each approach toward four different depths: 125×, 250×, 500× and 1,100×. The consistency of true positives and two different types of false positives (nonvariant and germline) using 39 truth sets are shown with colored parts representing the call set specific to each depth (nVennR). MH, MosaicHunter; DM, DeepMosaic; MT2-to, Mutect2 tumor-only mode; HC-p20, HC with ploidy option 20; HC-p200, HC with ploidy option 200; TP, true positive. Source data for this figure are provided.

mosaic mutations[11,12,20], (2) procedures previously used to discover mosaic variants[9,21] and (3) algorithms that can be adapted for mosaic mutation detection via simple usage modifications[10] (Methods). The 11 strategies were classified into four major categories based on their baseline algorithms: mosaic, somatic, germline and ensemble (Extended Data Fig. 1). The mosaic category includes three algorithms designed to specifically target mosaic mutations: MosaicHunter (MH)[20], MosaicForecast (MF)[11] and DeepMosaic (DM)[12], which exploit Bayesian, Random-Forest and image-based deep-learning algorithms, respectively. The somatic and germline categories include the three most frequently used variant callers Mutect2 (ref. 22) (MT2), Strelka2 (ref. 23) (STK2) and HaplotypeCaller[24] (HC), with or without previously applied modifications. Modified filtrations were applied to MT2 to allow mutant alleles in control samples for shared variant calling[10] (Methods). For HC, modified ploidy assumptions (ploidy 20, HC-p20

and ploidy 200, HC-p200) were applied to detect low- to medium-level mosaic mutations based on the most recent recommendation[9] (setting ploidy to the 20% of the overall sequencing coverage) (see Supplementary Notes for other options). There was only one ensemble approach, M2S2MH, that integrates three different callers (MH, MT2 and STK2) with additional read-level filtration[21].

## Evaluation of single-sample-based calling

This task aims to sort out true mosaic variants from sequencing artifacts and germline variants using one sample (without a matched control) (Fig. 2a). Most approaches primarily target the variants within intermediate VAFs (5–35%) that are not likely acquired by sequencing artifacts (less than 1–5%) or germline variants (50–100%). We evaluated six currently applicable approaches (MH, MF, DM, MT2-to (MT2 tumor-only mode), HC-p20 and HC-p200) for mosaic SNV and four

approaches (MF, MT2-to, HC-p20 and HC-p200) for mosaic INDEL detection (Methods).

For mosaic SNVs, MF and MT2-to showed the best performance in low to medium VAF ranges (4–25%); MT2-to was higher in sensitivity and lower in precision than MF (Fig. 2b). Most of the performance gain in MF and MT2-to was achieved at low VAFs (less than 10%) supported by high sensitivity, whereas the absolute number of false positives was high (false positives per 1 Mbp = 0.69 in MT2-to, 0.71 in MF and 0–0.29 in the others). In the high-VAF area (more than or equal to 25%), MF and MH performed best in terms of F1 score, whereas MF and HC-p200 were best in terms of the area under the precision-recall curve (AUPRC) (Fig. 2b). This indicates that HC has a margin for improvement by parameter optimization in high-VAF mosaic SNV calling. For INDELs, MF showed the best performance for all VAF ranges in terms of the F1 score, but the overall accuracy was lower than that of SNVs (Fig. 2c). No current algorithms could efficiently detect INDELs of a very low (less than 5%) VAF even at an ultra-high (1,100×) depth. Similar to that in SNVs, HC-p20 and HC-p200 marked the best AUPRCs at medium to high-VAF ranges (more than or equal to 16%). We further found that MF and MT2-to also showed a good balance between precision and sensitivity ($\log_2$ ratio = −0.31 and −0.78 in MF and MT2-to), whereas the other tools were more biased toward precision (DM = 1.8, MH = 3.08 and HC = 0.53–4.91) (Fig. 2d).

Comparisons among the different call sets identified the overall discordance between the algorithms (Fig. 2e). Except between MT2-to and MF or HC-p20 and HC-p200, no two sets of called mosaic variants agreed more than 32% (percentage agreement 8–32, Fig. 2e left). Moreover, the agreement was far lower in false positive calls, forming a distinctive error set for each algorithm, mostly in low VAFs (less than 10%) (Fig. 2e right, Extended Data Fig. 2). This tendency was shown in all sequencing depths tested (Supplementary Fig. 1a,b). By these results, the following conjectures about the combinatorial use of multiple callers could be inferred: (1) each algorithm finds distinct parts of the true mosaic variants and can be complemented by each other (Supplementary Fig. 1c), (2) most of the false positive calls from one algorithm can be successfully filtered in others, but (3) simple ways to use two or more callers (for example, taking intersection) would substantially lower accuracy. We tackle this problem more thoroughly later in this study to compose a better ensemble approach in single-sample analysis than those generally used so far.

We further investigated the advantage of sequencing depth on performance. As expected, sensitivity and precision were improved in most approaches with the increase in sequencing depth, for both SNVs and INDELs (Supplementary Fig. 2). For example, the F1 score of MF was 0.20, 0.38, 0.54 and 0.62 at 125×, 250×, 500× and 1,000×, respectively. The only exception was DM, which marked the best performance at 250×, where the algorithm was trained[12]. It indicates that diversifying sequencing depth in training data is encouraged for deep-learning algorithms. Conversely, we found an unexpectedly large amount of depth-specific true and false positive calls in all algorithms (Fig. 2f and Supplementary Fig. 1d for INDEL). This observation indicates that the use of high-depth sequencing has not been optimized for mosaic variant detection. Because the loss of (previously detected) true calls and the gain of (previously eliminated) false calls in a higher depth are algorithmically undesirable, future efforts should be made to take advantage of high-depth sequencing data properly.

**Evaluation of paired-sample-based calling**

Multi-sampling is a growing option for optimal mosaic variant profiling. Here, we evaluate mosaic variant calling from a paired (two) sample, either one or both of which are affected to form sample-specific or shared variants, respectively (Fig. 3a). Detection of a sample-specific mosaic variant is equivalent to the conventional somatic variant detection problem. By contrast, detecting shared variants is more challenging because of the increased uncertainty about variant compositions.

For example, VAFs of mosaic variants can be substantially varied or absent within samples depending on their developmental lineages, local proliferation and selective pressure across tissues[1]. While no single variant caller has tried to detect shared mosaic variants, there have been a few attempts to apply existing algorithms with modified usages[10,21].

We evaluated 11 and eight strategies that could be applied to detect mosaic SNVs and INDELs in paired samples, respectively. These strategies can be divided into two major categories: single genotyping and joint genotyping (Fig. 3b). Single genotyping initially calls variants in each sample independently, and then finds mosaic variants by combining the two call sets (Methods). We applied MH, MF, DM, MT2-to, HC-p20 and HC-p200 to the single-genotyping approaches for SNVs and MF, MT2-to, HC-p20 and HC-p200 for INDELs. The joint-genotyping category considers both samples together at the same time to call sample-specific and/or shared mosaic variants. MT2 and STK2 were eligible for detecting sample-specific mosaic SNVs and INDELs. Modified MH paired mode (MHP-m; Methods) were applicable to detect shared SNVs only, whereas Modified MT2 paired mode (MT2-m; Methods) were applicable to detect both shared SNVs and INDELs. M2S2MH was the only strategy that directly targeted the shared mosaic SNVs per se along with sample-specific SNVs and INDELs (Methods).

In sample-specific mosaic SNV detection, three of the joint-genotyping methods, MT2, STK2 and M2S2MH, showed best accuracy (Fig. 3c,d) throughout all sequencing depths (Supplementary Figs. 3 and 4). In particular, higher sensitivity at low VAF (less than 10%) (0.72, 0.72 and 0.70 in MT2, STK2 and M2S2MH, respectively) than the others (0.01–0.06) was observed (Fig. 3e and Supplementary Fig. 4). These tools also marked higher precision at low VAFs (MT2 = 0.58, M2S2MH = 0.70, STK2 = 0.4) than single-genotyping methods (MF = 0.33 and MT2-to = 0.37), suggesting the benefit of joint genotyping. In sample-specific INDEL detection, MT2 and STK2 marked the best accuracy in low/high (less than 5% and greater than 25%), and intermediate VAFs (5–25%), respectively. M2S2MH showed almost perfect precision with lower sensitivity, which can be useful for prioritizing call sets (Supplementary Figs. 5 and 6).

In shared mosaic variant detection, MF showed the best overall F1 score (MF = 0.92) (Fig. 3c,d, data points in the plane). However, the performance was largely variable to conditions, especially to VAFs (Extended Data Fig. 3). Partitioning the VAF space of a sample pair into 16 (= 4 × 4) areas depicted the landscape of the local best performers (Fig. 3f,g). For SNVs, MT2-to marked the best F1 score in nine areas, mostly of intermediate VAFs (5–25%). MF performed best in two areas where variants in both samples are in low (less than 5%) and high (greater than 25%) VAFs, the former of which is frequently the main target in current mosaicism studies. In three areas of higher VAFs (greater than 25%), germline approaches (HC-p20 and HC-p200) showed the best performance, especially with high sensitivity. For INDELs, with the more complex pattern of the best-performing callers, MT2-to, MF and HC-p20/p200 generally outperformed others as in SNV detection (Fig. 3g). Using the identified VAF-specificity, we constructed and tested an instant ensemble strategy that selects call sets only from the local best performers to find a substantial improvement in overall F1 score for SNVs (0.89 to 0.96) and INDELs (0.52 to 0.60) from the best single callers, which we expect to direct future development.

Finally, we analyzed the effect of VAF imbalance (that is, the same variant is presented in different VAFs) in the detection of shared mosaic variants (Fig. 3h). Most algorithms showed reduced sensitivity for unbalanced (more than or equal to twofold VAF difference) mosaic variants. However, we found that MHP-m, MT2-m and M2S2MH were robust to the imbalance, demonstrating the effectiveness of joint genotyping. Another problem that VAF imbalance causes is the misclassification of shared variants as sample-specific variants, which happens by missing calls in one sample (Fig. 3i). We found that HC-p20, HC-p200, MH and DM lost up to twice the number of called true positives by misclassification, posing an opportunity to improve sensitivity

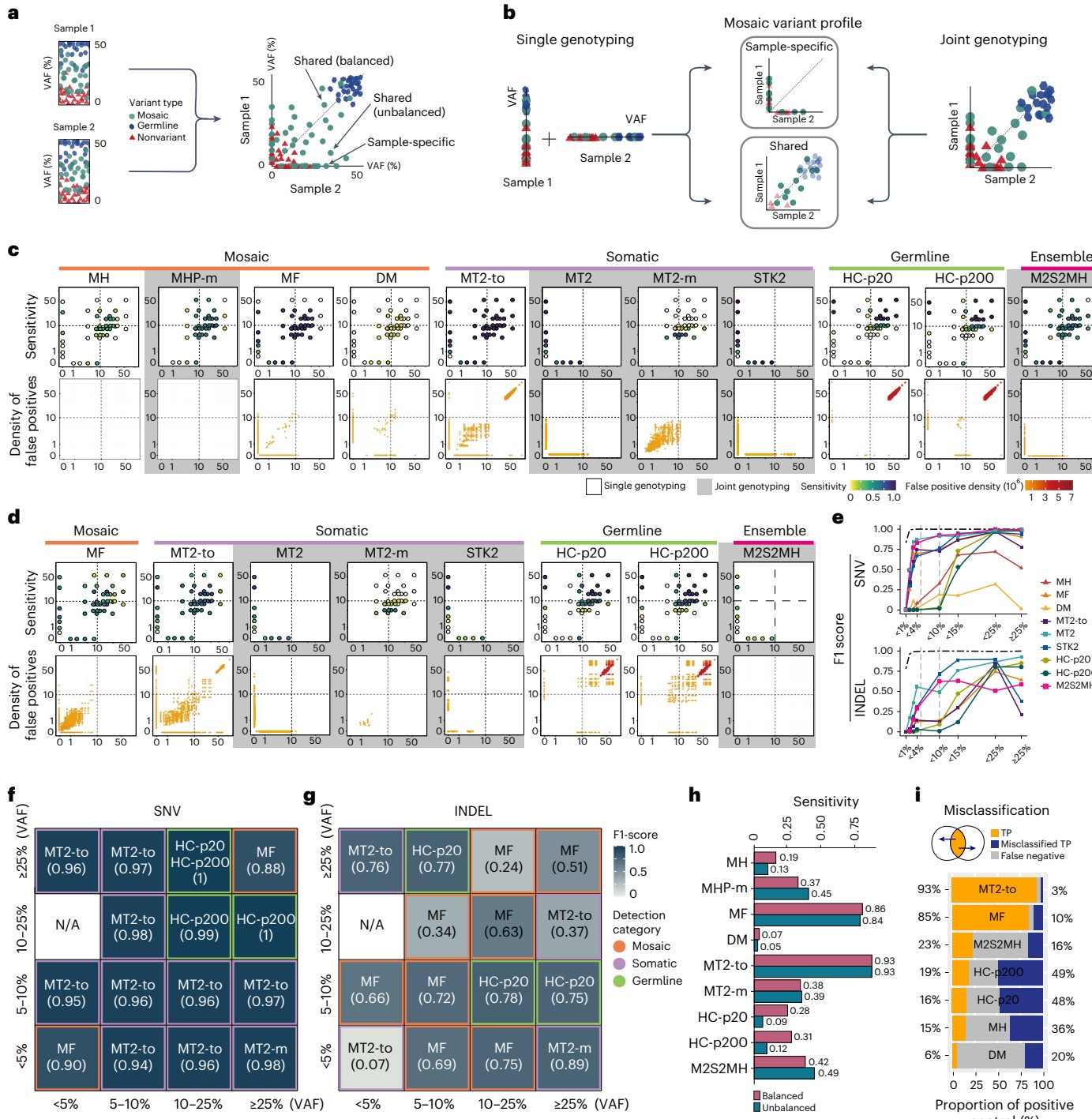

**Fig. 3 | Evaluation of paired-sample-based mosaic variant calling.**
**a**, A schematic overview of detection of shared or sample-specific variants in a paired sample. **b**, Single and joint genotyping described to detect shared and sample-specific variants. Whereas single genotyping detects variants for each sample and take intersections or differences of the call set, joint genotyping takes both samples simultaneously. **c,d**, Sensitivity and false positive distribution of the 11 approaches in 1,100× for shared SNV (**c**) and eight approaches for INDEL detection (**d**). Distribution is shown across $\log_{10}$ scaled VAFs on each axis. Gray box denotes for joint genotyping and dashed cross lines depict VAF 10% for both axes. **e**, F1 scores of detecting sample-specific SNVs and INDELs are shown with two dashed lines point to 5 and 10% VAFs. Performance is shown in seven VAF bins (<1, 1–2, 2–3, 3–4, 4–10, 10–15, 15–25 and ≥25%). Dashed line with dots depicts the

theoretical upper limit for F1 scores. **f,g**, Partitioned F1 scores of shared mosaic SNVs were calculated in 16 areas with the combinations of the four VAF range groups (very low ≤5%, low >5 and ≤10%, medium >10 and ≤25%, and high >25%). NA, not evaluated (could not assign positive controls). Combinations of the best-performing approaches within each combinational VAF area are shown for shared SNVs (**f**) and INDELs (**g**) in 1,100× data with 39 truth sets. **h**, Sensitivity comparison with 1,100× data between the balanced and unbalanced VAF combinations in shared mosaic variant detection. The unbalanced category includes variants whose differences in VAF of two samples was greater than twofold. **i**, Proportion of misclassified shared variants to sample-specific variants in 1,100× data (39 truth sets) are shown with the proportion of true positives. Source data for this figure are provided.

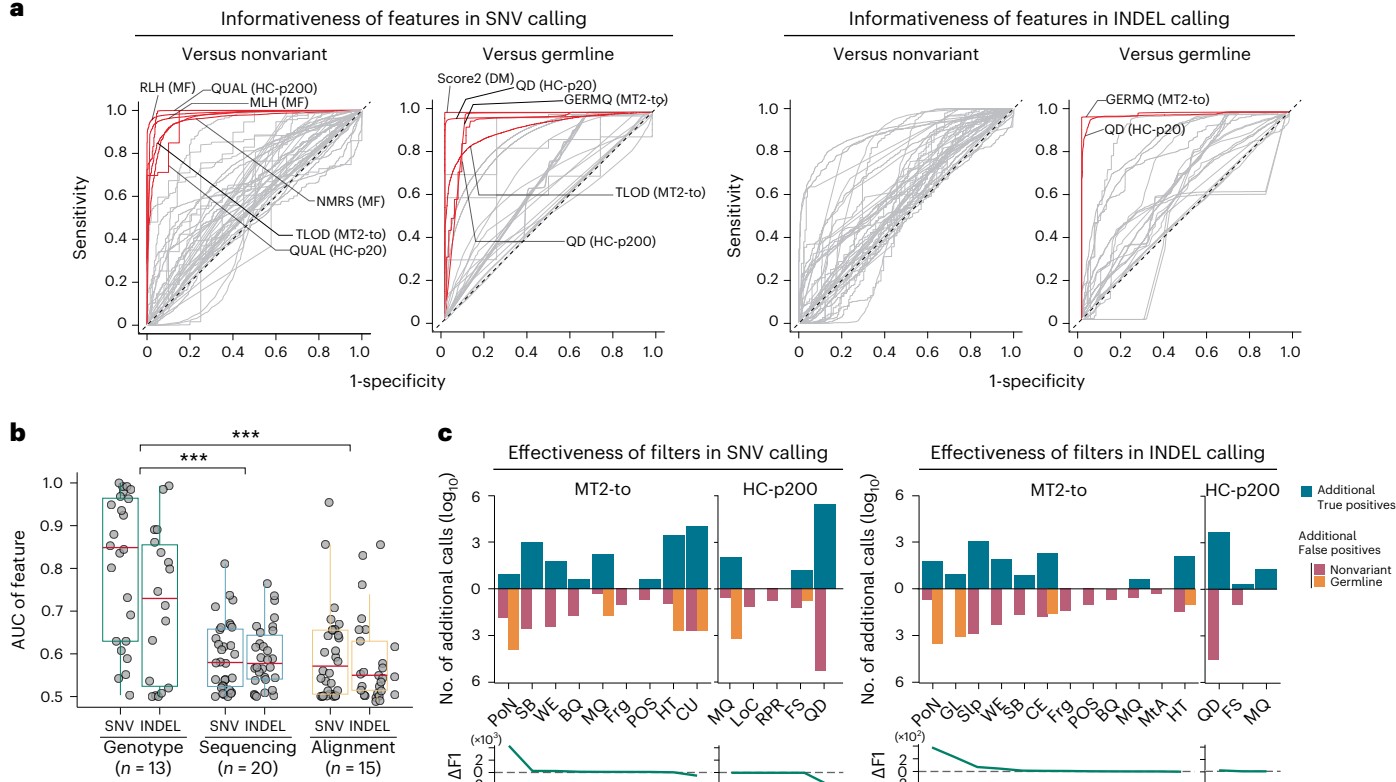

**Fig. 4 | Evaluation of the features and filters. a**, Receiver operating characteristics of used features to distinguish two types of false positive (nonvariant and germline) from true positives are shown. Forty-eight features used in the four approaches (MF, MT2-to, DM and HC) for mosaic SNV (left) and INDEL (right) detection are shown. See Supplementary Table 2 for the full names and definitions of the evaluated features. Features with AUC above 0.9 are presented in red lines. **b**, Grouped AUCs of the 48 evaluated features in three categories: sequencing, alignment and genotype-level category. The median of each category in red lines, first and third quantiles with boxes, and minima and maxima with whiskers. Two-sided Wilcoxon rank sum test, $P = 3 \times 10^{-5}$ and

$2 \times 10^{-5}$ for the sequencing- and alignment-level features. **c**, The efficiencies of 16 postfilters from MT2-to and HC-p200. The number of true and false positive counts are shown in the $\log_{10}$ scale (upper). The resultant differences in F1 scores (lower), Delta F1 scores, was calculated by F1 score in default settings subtracted by the F1 score with disabled-filter. All results were generated with 1,100× data from 39 truth sets. RLH Refhom-likelihood, MLH mosaic-likelihood, NMRS mismatches p, PoN panel of normal, SB strand bias, WE weak evidence, BQ base quality, MQ map quality, Frg fragment, POS position, HT haplotype, CE clustered events, GL germline, Slp slippage, MtA multiallelic, LoC low coverage, RPR ReadPosRankSum. Source data for this figure are provided.

---

without core-level modifications. Overall, regardless of the current performance, joint genotyping, expansion of VAF ranges and modeling for diverse sequencing depths (Supplementary Fig. 7b) can be the most feasible ways for the best detection of shared mosaic variants.

### Evaluation of features and filters

Variant calling uses multiple features inferred from sequencing data, such as simple field values in raw data (for example, base-call quality) or specifically formulated measures (for example, the proportion of clipped alt-reads with more than or equal to 10 bp). Because variant calling is a decision process that selects, calculates and organizes such features, feature-level evaluation provides fundamental resources for developers for accurate detection.

Using positive and negative calls, we evaluated 48 features used in four mosaic detection algorithms (MF, MT2-to, DM and HC; see Supplementary Table 4 for the full list of features) for their classification powers on current false calls. The area under the curve (AUC) of the 48 features widely ranged from 0.5 to 1, 11 of which marked AUC > 0.9 (Fig. 4a). In particular, 'refhom-likelihood' (AUC = 0.99) and 'mosaic-likelihood' (AUC = 0.98), the likelihoods for wide-type and mosaic genotypes from MF and 'QUAL' (AUC = 0.99), a key confidence feature for genotyping in HC, were able to further discriminate nonvariant errors. Likewise, 'score 2' (potential mosaic possibility, AUC = 0.99) of DM and quality by depth (AUC = 0.98) of HC could

be used for filtering germline false positives. For INDELs, 'GERMQ' (Phred-scaled quality that alternative alleles are not germline variants, AUC = 0.99) of MT2-to and quality by depth (AUC = 0.98) were features that can distinguish germline false positives (Fig. 4a, right). There was no prominent potential feature for identifying true INDELs from nonvariant artifacts.

We inspected the properties of the informative features (Fig. 4b). The 48 features could be grouped into three categories by their source of origin: (1) sequencing ($n = 20$), (2) alignment ($n = 15$) and (3) genotype ($n = 13$). Sequencing-level features are the raw values and their derivatives regarding sequencing reads, base-call and quality. Alignment-level features are values that annotate the patterns and noises in sequencing read mapping, and genotype-level features are the intermediate scores generated during genotyping, such as likelihoods and confidence values for a position to be a true variant. We found that most of the informative features (12 of 13, 92%) fell into the genotype level. Additionally, the overall AUC of genotype-level features was significantly higher than those of sequencing- and alignment-level features (Wilcoxon rank sum test $P = 3 \times 10^{-5}$ and $2 \times 10^{-5}$, respectively, two-sided) (Fig. 4b), suggesting that a better and more active use of genotype-level features would benefit developers when constructing a new model for mosaic variant detection.

Next, we evaluated another usage of such information: a postfilter with which the call sets are postprocessed and refined by using a single

threshold value. A good filter is expected to remove the false calls, while leaving the true calls, leading to an improvement in overall accuracy (for example, F1 score). We tested the efficiency of the 16 independently adjustable filters used by MT2-to and HC-p200 by disabling them and comparing the differences in the call sets (Fig. 4c, upper). We found that most filters removed a substantial number of false positives (0 to 233,874); however, it was also accompanied by the corresponding number of lost true positives (0 to 160,967). Overall, the contribution of the filters to the overall performance (F1 score) was strictly limited (−0.002 to 0.038, mean 0.003) (Fig. 4c lower), demanding more advanced strategies (for example, machine-learning-based scoring) for future improvement.

### Potential strategies for improving accuracy

The complexity of algorithms and numerous variables therein require sophisticated usages of data and optimization of strategies for improving mosaic variant calling. Here, we formulate three strategies that can be used for future development and demonstrate their feasibility.

First, we tested the feature-level recombination of multiple algorithms, as an extension of the call set-level integration that brought an instant improvement in the paired-sample-based detection (Fig. 3f,g). Here, informative features from one algorithm are imported into other algorithms, in pursuit of a synergistic or complementary effect. For example, the low precision of MT2-to at the low VAF (lower than 10%) in the single-sample detection can be reinforced by foreign features that discriminate nonvariants. A simple implementation proved that augmenting 'alt softclip' (of MF) to MT2-to could remove 27.3% (228 of 834) of the false calls while only losing 0.009% (27 of 289,124) of the true answers (Extended Data Fig. 4a top). Likewise, applying 'MFRL alt' (of MT2-to) (a median fragment length of reads with alternative alleles) or 'Het Likelihood' (of MF) (genotype likelihood of the variant being germline heterozygous) to the HC-p200 call set removed 40.54% (30 of 74) and 93.08% (50,304 of 54,043) of the false calls from the nonvariant and germline sites, respectively, losing 0.03% (26 of 86,415) and 3.3% (2,732 of 82,026) of the true calls, respectively (Extended Data Fig. 4a middle and bottom).

Second, we tested the broader usage of the 'rescue' procedure (of M2S2MH), re-evaluation of positions that were only called from the matched sample but not in the other (Extended Data Fig. 4b and Methods). We confirmed that the procedure increased the F1 score by 2.5-fold (0.26 to 0.64) from the original MH calls in the M2S2MH algorithm (Extended Data Fig. 4c). Applying it to the two different callers MF and DM also achieved an increase in the F1 score by 0.1 and 0.26, respectively (Extended Data Fig. 4c). This result indicates that the rescue strategy can be used as an efficient post process for high-precision algorithms to reinforce the sensitivity of their models.

Finally, we formulated an approach that uses developmental hierarchy within many (more than or equal to three) samples for enhanced precision in mosaic variant calling in which no such algorithms have been developed yet. We assume two samples that are originated from more recently differentiated tissues as more proximal in a developmental lineage tree, such as brain and spinal cord. Then, variants that are not shared between these two tissues are not likely to be present in a more developmentally distal third tissue (for example, heart) (Extended Data Fig. 4d and Supplementary Notes). This relationship can be used to filter out false mosaic variants. A pilot application of this approach was conducted on the 1,944 possible combinations of three different mosaic samples (out of 39 mixtures), by selecting one from three different mosaic types (nine in M1, 12 in M2 and 18 in M3) from the truth set. We found that 67.3 and 55.5% of the shared false positives of SNVs and INDELs could be removed while losing only a small fraction (1.6 and 3.9%) of true positives, thereby increasing the F1 scores from 0.94 to 0.95 and from 0.18 to 0.29 for SNVs and INDELs, respectively (Extended Data Fig. 4e). Therefore, we anticipate that a generalized algorithm that takes advantage of joint genotyping and multiple tissues would increase the sensitivity and precision of shared mosaic variant calling.

## Discussion

In this study, we presented a systematic benchmark of mosaic variant calling strategies. Our analysis revealed the sequencing depth-, VAF-, variant-sharing pattern- and error type-specific strengths and diversity of the current algorithms, feasible strategies for ensemble approaches and directions for future development. After all the detailed analyses, we summarized the conclusions from the benchmark results in Fig. 5 and practical recommendations for users and developers in Table 1. The applicability and reproducibility of these recommendations were confirmed in three independent biological datasets[9,10,21] (Supplementary Notes).

The technologies used for detecting germline and somatic mutations have been greatly improved in recent decades. The F1 score of germline variant calling has exceeded 0.99 (ref. 18). Clonal somatic mutation calling (for example, cancer) has gained substantial credence on a clinical level, by lowering the limit of detection down to around 1% (ref. 25). Nevertheless, the accuracy for mosaic variant detection was shown to be much lower in our analysis, especially for low VAF (less than 5%) SNVs (F1 < 0.74) and INDELs (F1 < 0.3) even with ultra-deep sequencing (1,100×) data. As this field is still in its infancy, efforts to increase its detection accuracy to the level of somatic mutations should be made. Again, proper evaluation of detection performance will expedite the development of this field and suggest its direction, as several benchmarks and competitions have led to the best practices for germline and somatic mutations[14,26].

The construction of robust and comprehensive reference standards is the critical prerequisite for benchmarking, as it was for germline (for example, Genome in a Bottle[15]) and somatic variants[17]. Additionally, genome editing technologies (for example, CRISPR–Cas9) have been applied to spike-in somatic variants of designated allele fractions to produce commercial products[27]. Generating in silico simulated datasets is also a simple but powerful method, such as the BAM-file mixing[17]. However, we noted that mixing BAM files greatly limits the variety of noises (for example, sequencing errors) to the source data, and generates much simpler error profiles (Supplementary Fig. 11 and Supplementary Notes). We believe that our approach to constructing a mixture-derived standard reference is an appropriate way for securing both the scale and robustness for mosaic variant analysis.

In this benchmark, we focused on the conventional bulk sequencing used by most researchers. However, further sequencing technologies for detecting low-VAF (less than 5%) variants are being developed, and existing ones have proven to be effective, such as linked-read sequencing[28,29] and unique molecular identifiers (UMI)[30]. Continuing efforts should be made on the proper evaluation of these methods and their applications to detect mosaic variants.

Despite all efforts, this benchmark has potential pitfalls, particularly in the interpretation of the analysis results, limitations in the search space and data dependency. First, the performance of the mosaic calling 'strategies' should not be confused with their baseline algorithms, especially when they were used in an unintended way. For example, the performance of MT2-m does not directly indicate the somatic mutation calling performance of MT2 with the modified use of normal filters. Likewise, MHP-m originally reported variants with inconsistent genotypes in two samples (for example, germline in one and mosaic in the other) and has been modified to call the shared variants by referring to the internal genotype probability matrix. The compositions of all the single-genotyping approaches for shared variant detection (MH, MF, DM, MT2-to and HC-p20/p200) are generally acceptable; however, the usages were not explicitly declared in the original algorithms. Again, these modifications were conducted to test the potentially applicable strategies in the absence of specifically developed algorithms. We also would like to note that the complexity of

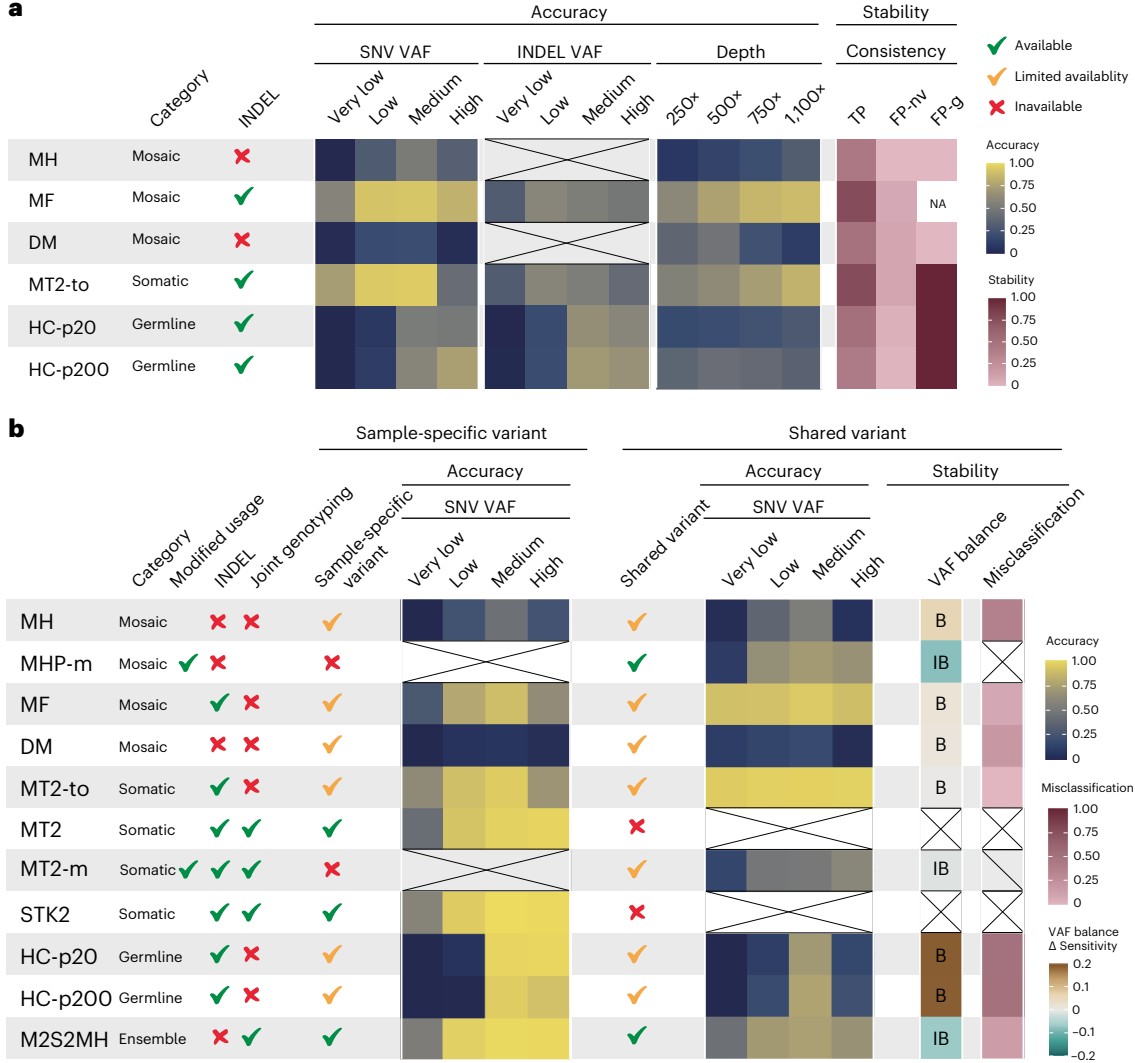

**Fig. 5 | Summary of the benchmark. a,** A benchmark summary of mosaic variant detection in single sample. AUPRC of the detection strategies are shown within four VAF ranges (very low ≤5%, low >5 and ≤10%, medium >10 and ≤25%, and high >25%). The median value of the AUPRC within the four VAF ranges was used for the representatives for each depth in SNV detection (125×, 250×, 500× and 1,100×). Call set consistency is shown in three different types of call set: true positives (TP), nonvariant false positives (FP-nv) and germline variant false positives (FP-g). **b,** Benchmark summary of mosaic variant detection in a

paired sample is shown. Properties of each strategy are marked with a green tick, yellow tick and red cross, denoting available, limitedly available and unavailable, respectively. VAF ranges are same as noted in **a.** VAF balance was calculated by subtracting the sensitivities of balanced from unbalanced VAF pairs. B (balance) or IB (imbalance) were noted within the boxes as one with the larger sensitivity among them. Misclassification refers to the proportion of misclassified shared variants to sample-specific calls out of the total positive controls in each strategy. Source data for this figure are provided.

the use of parameters limits the benchmark. At least 4–110 parameters can be used and adjusted, and the number of combinations reaches up to tens of millions. Because of the intractability, evaluations have been conducted using default parameters, assuming that the empirical suggestions from developers (default parameters) are close to the optimal.

Second, our benchmark focuses on capture-based deep exome sequencing that have distinct characteristics from other technologies (for example, WGS and panel-seq). Capture-based exome sequencing accompanies read-depth variability within and across the exon region, especially for INDELs. Even though securing high sequencing depths with exome data can greatly advantage the detection accuracy, the low-VAF (less than 10%) variant detection accuracy remains challenging due to the coverage unevenness (Supplementary Notes). Thereby, we expect similar or slightly higher performance with whole genome- or panel-sequencing data of the same target sequencing depths (Supplementary Notes). Furthermore, genomic regions of low complexity

(for example, repeats and segmental duplications) were excluded in this benchmark (Methods). We are aware of that these low complexity regions are also of interest for researchers and developers. However, we focused on the high-confidence regions to evaluate the core performance of the tools, as empirical or arbitrary filtration strategies are more actively used in the low complexity regions.

Finally, the composition of the standard reference can affect the results, such as the distribution of VAFs in the datasets, number of positive controls, sequencing platform, read length and error profiles. First, the cumulative cell line mixing (Fig. 1) produced a large set of robust mosaic variants, but it could also alter the composition of germline variant sites. We resolved the issue of loss of pure germline sites by preparing a separate set of reference standards that recovered germline sites[19]. Second, we noted that germline variants from multiple cell lines may be presented as somatic-like variants of low to medium VAF after mixing. This increases the number of extra

**Table 1 | Recommendations for mosaic variant detection**

| | | Recommendation |
|---|---|---|
| **Fundamental concept** | 1 | Mosaic variant calling is not a task of a single kind, but a set of diverse problems, whose characteristics and difficulties vary largely. The number of samples, presence of matched controls, VAFs and variant-sharing patterns are major factors that affect the algorithmic performance and should be carefully considered before analysis. |
| **Single-sample detection** | 2 | MF and MT2 are generally recommended for calling mosaic SNVs in a single sample. These tools are particularly strong in calling low-VAF mutations (less than 10%). |
| | 3 | MH showed low sensitivity but extremely high precision. Together with other callers, these calls can be used for strict filtering or prioritization. |
| | 4 | MF is considered the current best algorithm for mosaic INDEL calling, whereas no algorithms are successful in calling low-VAF (less than 4%) INDELs. |
| | 5 | Call set concordance between callers and read depths is very low. Therefore, finding overlaps from different callers to achieve high confidence is not recommended. The use of multiple callers should be composed in a way of assigning one to the best-performing VAF area. |
| **Paired-sample detection** | 6 | For sample-specific calling, current somatic callers (MT2 and STK2) outperform in overall VAFs including low VAF (less than 5%). |
| | 7 | For shared variant detection, although MF and MT2 showed good overall accuracy, we want to highlight that the best-performing tool for each VAF range varied. We recommend also trying out the best tools for the expected VAF-pair area if one exists (Fig. 3f). |
| | 8 | In the present situation, we recommend using somatic callers (MT2 or STK2) for sample-specific and MF with the rescue strategy for shared mosaic variant detection with paired samples. |
| | 9 | Although no specific algorithms explicitly handle more than two samples, using sample proximity in terms of developmental lineage can be an efficient filtration strategy. |
| **For method developers** | 10 | Specify the target problem first. What types of mosaic variant are considered? Is matched control sample required? Are the variants shared in multiple samples? The final performance can be completely different when applied to problems of unintended forms. |
| | 11 | We recommend considering various sequencing platforms or read depths in algorithm development. Algorithms that exploit machine-learning techniques (for example, deep learning) can be further improved by varying the training sets especially in terms of VAF and sequencing depth. |
| | 12 | Provided with numerous features, scores and filters from multiple algorithms, a complementary ensemble approach can be an efficient start to improve performance without developing a new algorithm from scratch. |

variants in a single read (Supplementary Notes) and may negatively affect the performance of tools that consider genotypes of flanking sites including MT2, although we confirmed the effect is none or small. Third, the use of germline variants as origin may affect the single-nucleotide polymorphism (SNP)-based postfiltration processes (filtration of SNP sites after or during the calling process) that MH, DM and MT2 exploit. We removed the true variant positions from external population frequency databases such as dbSNP, gnomAD and panel of normal that are fed into the tools, and confirmed the issue has been successfully resolved (Supplementary Notes). However, the modification of the population frequency databases may also enhance sensitivity by invalidating the filtration that may act negatively for mosaic variants that hit SNP sites by chance. We also confirmed that the actual effect is minor (Supplementary Notes) but should be noticed. Last, due to the large number of control positives, we not only recalibrated the precision and F1 score, but also provided an absolute measure, false positive rate (FP per 1 Mbp). All benchmark results should be carefully explained because those measures involve relativity in interpretation.

In summary, we anticipate that our study will guide researchers to the best use of current algorithms. We expect that accurate analysis of mosaic variants will broaden our understanding of fundamental human development, as well as other model organisms[5,31]. Also, for method developers, our study will be a good starting point for technical advances in mosaic variant calling to the germline and somatic variant levels.

## Online content

Any methods, additional references, Nature Portfolio reporting summaries, source data, extended data, supplementary information, acknowledgements, peer review information; details of author contributions

and competing interests; and statements of data and code availability are available at https://doi.org/10.1038/s41592-023-02043-2.

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

## Methods

### Datasets

We obtained a set of mosaic reference standards (BAM) based on the accumulative mixing of pregenotyped normal cell lines from the Sequence Read Archive (PRJNA758606). The integrity of the used cell lines was confirmed with short tandem repeats comparison and verified to have no mycoplasma contamination (Supplementary Notes). Diverse combinations and ratios of the mixtures generated 39 reference materials, harboring both abundant mosaic variants of VAF (Supplementary Table 1) and two types of negative control: nonvariant sites (Set A) and germline variants (Set B)[19]. The 39 reference materials were included in one of the three mixture categories (nine M1, 12 M2 and 18 M3) that represent the genotypes of distinct lineages. Among the full sets of controls in every reference material, we only considered high-confident regions by excluding the simple repeats and segmental duplications supplied by UCSC (University of California Santa Cruz). Thus, 345,552 positive SNVs (92%) and 8,706 INDELs (72%) were adjustable, and each of the 39 reference materials contained two types of negative control: 33,093,574 nonvariant (94%) and 18,151 germline sites (91%) per sample. Information on the positive and negative controls used here is shown in Supplementary Table 1. We also performed multistep down-samplings (125×, 250× and 500×) of original reference materials (1,100× on average) to comprehensively assess the performance under different sequencing depths.

### Variant calling

Eleven detection approaches were applied to the truth sets. For the single-sample and single-genotyping analyses, MH (v.1.0, single mode); MF (v.0.0.1, 250× trained models for SNVs); DM (v.0.0, efficientnet-b4_epoch_6.pt); MT2-to (v.4.1.9.0, tumor-only mode and applied FilterMutectCalls); GATK HC (v.4.1.8.0) with a ploidy option of 20 and 200 (see Supplementary Notes for the ploidy settings), with the quality filters, were adjusted according to the criteria based on GATK Variant Quality Score Recalibration (a quality by depth greater than or equal to 2, FS (FisherStrand) ≤ 60, DP (Depth) ≥ 20, map quality equal to or greater than 40, ReadPosRankSum ≥ −8 and −2.5 ≤ MQRankSum ≤ 2.5). Raw variant calls from MT2-to were used as input for MF and DM[11,12]. We also tried to evaluate MF with the raw calls of HC with ploidy 200, suggested by another study[9], but the MT2-to call set was observed to be more appropriate for evaluation (Supplementary Notes). Variants tagged as 'mosaic' (MH and MF) or 'PASS' (DM, HC and MT2-to) were only kept for downstream analysis. The population frequency data was needed for default settings, dbSNP(b154) for Mosaic Hunter and panel of normal (1000g_pon.hg38.vcf.gz) from GATK resource bundle (https://console.cloud.google.com/storage/browser/genomics-public-data/resources/broad/hg38/v0) for MT2, were applied after the removal of positive controls; this was done because the positive controls were attained from mutually exclusive germline variants[19]. Segmental duplication and simple repeats from UCSC were used to remove both the low-confident regions in the control sets and in the tools (MH, MF and DM) if used as a part of the algorithm[11,12,20]. For paired-sample analysis, STK2 and MT2 were applied with default parameters, and only PASS calls were used for downstream analyses. MT2-m and MHP-m were applied with simple modifications to detect the shared mosaic variants. Variants of MT2 paired mode tagged as 'normal artifacts' by FilterMutectCalls were selected for MT2-m to be exploited as an alternative filtering strategy for shared variant detection[10]. In MHP-m, paired naïve mode, if (1) the joint probability of two samples with 'mosaic' variants was greater than 0.05 and (2) if it was larger than that of any other genotype combination, the variants were considered as shared, whereas the remaining variants remained as sample-specific. For the M2S2MH approach, only concurrent calls of MT2 (paired mode) and STK2 (v.2.9.10, somatic) were exploited. Repeats and germline INDELs (with Manta[23] v.1.6.0) within five base pairs were removed for detecting sample-specific variants[21]. For shared variant detection, MH (single mode) was applied

to each sample, followed by a comprehensive filtering process using the read counts, depths and VAFs of both samples, as previously mentioned[21]. The detailed pipelines of the approaches used for all nine variant callings are shown in Extended Data Fig. 5.

### Performance evaluation of single-sample analysis

For each approach, the precision, sensitivity and F1 score were calculated based on the call set. To investigate the detection performance, the variants were divided into eight categories based on their VAF (less than 1, 1–2, 2–3, 3–4, 4–7.5, 7.5–9.6, 9.6–25 and equal to or more than 25%) for the SNVs and into seven categories (less than 1, 1–2, 2–4, 4–5, 5–9.6, 9.6–16 and equal to or more than 16%) for the INDELs so that almost equal numbers of variants were contained in each VAF category. Precision was recalibrated based on the density of positive controls (the number of positive controls per megabase), as a high positive control density overemphasizes the true positives over the false positives. The recalibration was done as follows with $w$, the weight of the overrepresented density of the positive controls:

$$w = \frac{\text{No. of expected positive controls}}{\text{No. of positive controls in the data}} \quad (1)$$

$$\text{precision} = \frac{\text{TP} \times w}{(\text{TP} \times w) + \text{FP}} \quad (2)$$

The number of expected positive controls was estimated based on the known previous densities of SNVs (1 per 1 Mb) and INDELs (0.1 per 1 Mb) (refs. 3,7), yielding $w$ values of 1/252.75 and 1/81.63, respectively. AUPRC was calculated with mosaic probability (MH and MF), score3 (DM), TLOD (MT2-to, MT2 and MT2-m), Somatic EVS (STK2) and quality by depth (HC-p20 and HC-p200).

Variant call set similarity between callers was calculated with Jaccard index in three different types: true positives, nonvariant false positives, and germline false positives. Call set consistency of four different sequencing depths (125×, 250×, 500× and 1,100×) was conducted with the R package nVennR[32] (v.0.2.3) by obtaining intersections and differences between call sets.

### Performance evaluation of paired-sample analysis

We selected a theoretical 'case' sample from one of 18 M3 mosaic types and one control out of 21 from the other mosaic types (nine from M1 and 12 from M2). Therefore, we tested 378 combinations of paired-sample analyses conducted with 1,481,274 shared variants (2,697 variants between M3 and M1 and 4,835 between M3 and M2). For single genotyping (MH, MF, DM, MT2-to, HC-p20 and HC-p200), case and control samples were applied independently, and their intersection and differences of the call sets were exploited for shared and sample-specific calls, respectively. MT2, STK2, MHP-m, MT2-m and M2S2MH were applied with their joint genotyping. For sensitivity comparisons between balanced and unbalanced VAF pairs, we referred to balanced shared variant if the difference in variant allele frequencies was less than twofold. Misclassified calls were collected if the shared variants (V1 for M3-M1 pair and V1 and V3 for M3-M2 pair) were only detected in either of each sample pair. Then, the proportions of true positives and misclassified calls were compared. For the F1 score comparison in VAF range combinations, we first divided the VAFs into four ranges of very low ≤5%, low >5 and ≤10%, medium >10 and ≤25%, and high >25%, to generate 16 shared VAF combinations to quantify the detection performances according to the VAFs. Among the 16 possible combinations, we could assign the shared variants of the reference standards to 15 groups, and by gathering the best F1 score for each combination, we could suggest an instant enhancement of shared variant detection using an ensemble of different detection approaches. The overall F1 score was calculated after normalization of the positive control counts in each VAF bin to ensure that the computed F1 score is not limited to this study.

### Evaluation of features and filters

To calculate the potential power of the features used in detection strategies, the AUCs of 48 features of MF, MT2-to, DM, HC-p20 and HC-p200 were obtained to see how well they further classify the true and false positives. True and false positives within the approaches were applied to pROC (v.1.17.01) and the AUC of each feature was determined. They were divided into three levels based on their intended use: 20 features into the sequencing level, 15 into the alignment-level and 13 into the genotype level. The full sets of evaluated features with the definitions, AUCs and categorizations are listed in Supplementary Table 4. Wilcoxon's rank sum test was used to compare AUCs of each group. To investigate filter efficiency, 16 postfilters from MT2 and HC were tested. We disabled each filter and compared the new F1 score to the original F1 score by collecting additional true and false positives. The difference in F1 score for each filter was calculated by subtracting the newly calculated F1 scores from the original F1 score. The postfilters of MH were excluded, as they were used in series; in other words, calculating their efficacy was highly dependent on the order of their application.

### Feature-level recombination

We tested whether a foreign feature of a distinct approach could be used in an independent variant call set from other approaches. We confirmed this hypothesis in three cases. Variants in the original call set could be tested when found in another approach accompanying the foreign features. Adjustable MT2-to calls (99% of true and 92% of false positives in nonvariant sites were adjustable) using the 'alt softclip' of MF (which was removed if the value was greater than 0.05) could filter out 27.3% of false positives with a 0.009% loss of true calls. Likewise, HC-p200 variant calls could be filtered using the 'MFRL alt' of MT2-to (less than 150, to 0.99% of true and 0.11% of adjustable false positives from nonvariant sites) and the 'Het likelihood' of MF (greater than 0.25, 95% of adjustable true and 8% of germline false positives).

### Lineage distance-based filtering

To demonstrate the advantage of using multi-samples (three, in this instance), we applied the shared variant call sets of MT2-to, of 1,944 combinations generated from nine M1, 12 M2 and 18 M3. We first collected the shared SNVs and INDELs between M1 and M2, which were more distal in lineage than M2 and M3. The variants were filtered out if they were not present in M3, and the resultant shared variants were compared to the original variant sets.

### Reporting summary

Further information on research design is available in the Nature Portfolio Reporting Summary linked to this article.

### Data availability

The data used for benchmark are available from the Sequence Read Archive under the accession code PRJNA758606 and the high-confidence ground truth sets used in this study are available in the GitHub repository (https://github.com/hiyoothere/Benchmarking-mosaic-variant-detection). Source data are provided with this paper.

### Code availability

The script used for evaluation is available in a public GitHub repository[33] (https://github.com/hiyoothere/Benchmarking-mosaic-variant-detection).

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

### Acknowledgements

S. Kim was supported by the Korea Health Technology R&D Project through the Korea Health Industry Development Institute (KHIDI) and Korea Dementia Research Center, funded by the Ministry of Health & Welfare and Ministry of Science and ICT (Information and Communication Technology), Republic of Korea (grant no. HU21C0286) and the Team Science Award of Yonsei University College of Medicine (grant award no. 6-2021-0007). Y-J.H. was supported by a grant of the Korea Health Technology R&D Project through the KHIDI, funded by the Ministry of Health & Welfare, Republic of Korea (grant no. HI19C1330). We thank Medical Illustration and Design, as a member of the Medical Research Support Services of Yonsei University College of Medicine, for providing excellent support with medical illustrations.

### Author contributions

S. Kim and Y.-J.H. designed and initiated the study. Y.-J.H., S. Kang, Jisoo Kim and Junhan Kim conducted the main analysis. Y.-J.H. and S. Kim wrote and edited the manuscript with input from coauthors. S.-Y.J. worked on data visualization. S. Kim led the project.

### Competing interests

S. Kim is cofounder of AIMA Inc., which seeks to develop techniques for early cancer diagnosis based on circulating tumor DNA. The other authors declare no competing interests.

### Additional information

**Extended data** is available for this paper at https://doi.org/10.1038/s41592-023-02043-2.

**Correspondence and requests for materials** should be addressed to Sangwoo Kim.

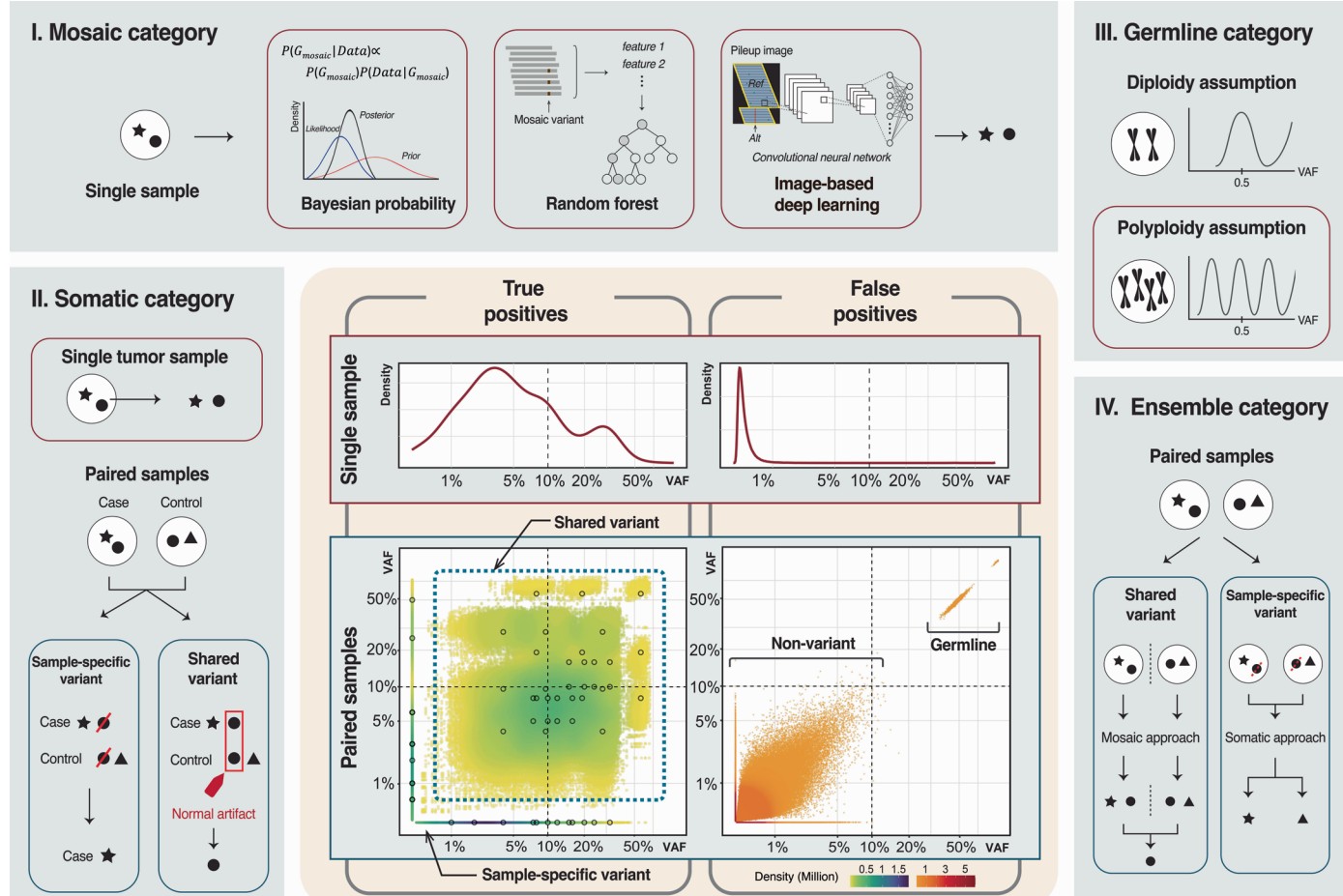

**Extended Data Fig. 1 | Landscape of truth set and mosaic detection methods in four categories.** Landscape of the true and false positives in the reference standards (center) and the applied detection approaches are shown (outer). Density of the variants along the VAFs in single and paired-sample analysis is shown in $\log_{10}$ scale in red and blue box respectively. The dashed line denotes VAF 10% and the data on the plane and axis refer to the shared (blue dashed rectangle) or sample-specific variants, respectively. Possible false positives from two types of negative controls (non-variant and germline variants) are shown after 1/1000 down sampling. Four categories of detection approaches based on their baseline algorithms, mosaic, somatic, germline, and ensemble categories.

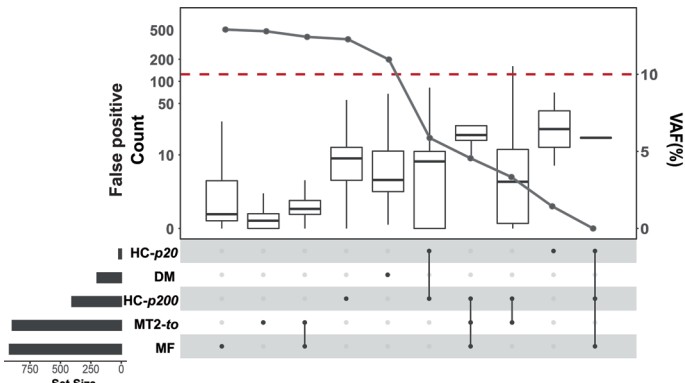

**Extended Data Fig. 2 | Compositions of the false positives from the evaluated detection approaches.** Intersections of the call sets and VAF distribution within each are shown with the left- and right-y axis respectively. Red dashed line refers to 10% of VAF and the median and quantiles are shown by the boxes with minima and maxima as whiskers. 39 truth sets in 1,100× depth were utilized for the analysis.

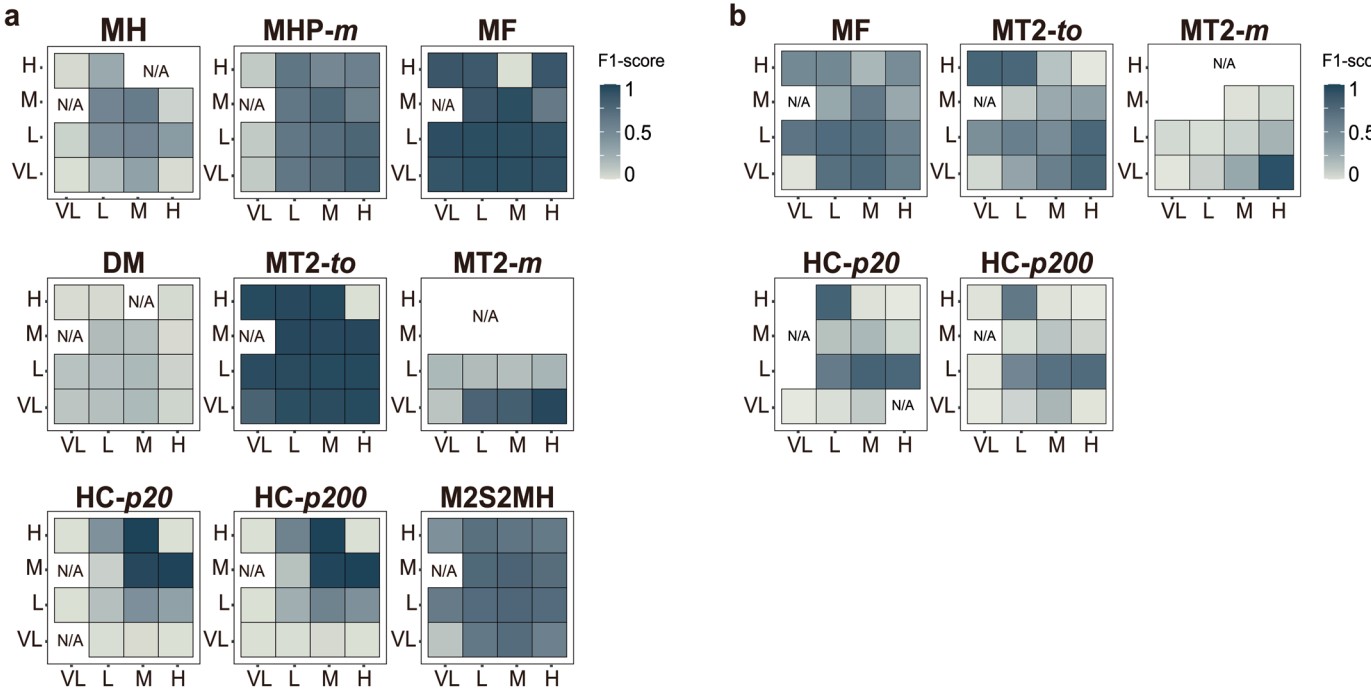

**Extended Data Fig. 3 | Evaluation on shared variant detection within VAF combinations. (a, b)** F1-scores of shared variant detection in sixteen VAF bin combinations are shown. 356 combinations of sample pairs were tested by taking eighteen M3 truth sets as the case sample and twenty-one M1 and M2 (9 M1 and 12 M2) sets as controls. Four VAF ranges (very low: ≤ 5%, low: > 5% and ≤ 10%, medium: > 10% and ≤ 25%, and high: ≥ 25%) were selected for analysis. VAF combination-specific F1 score of shared (a) SNV and (b) INDEL detection. VL very low, L low, M medium, H high.

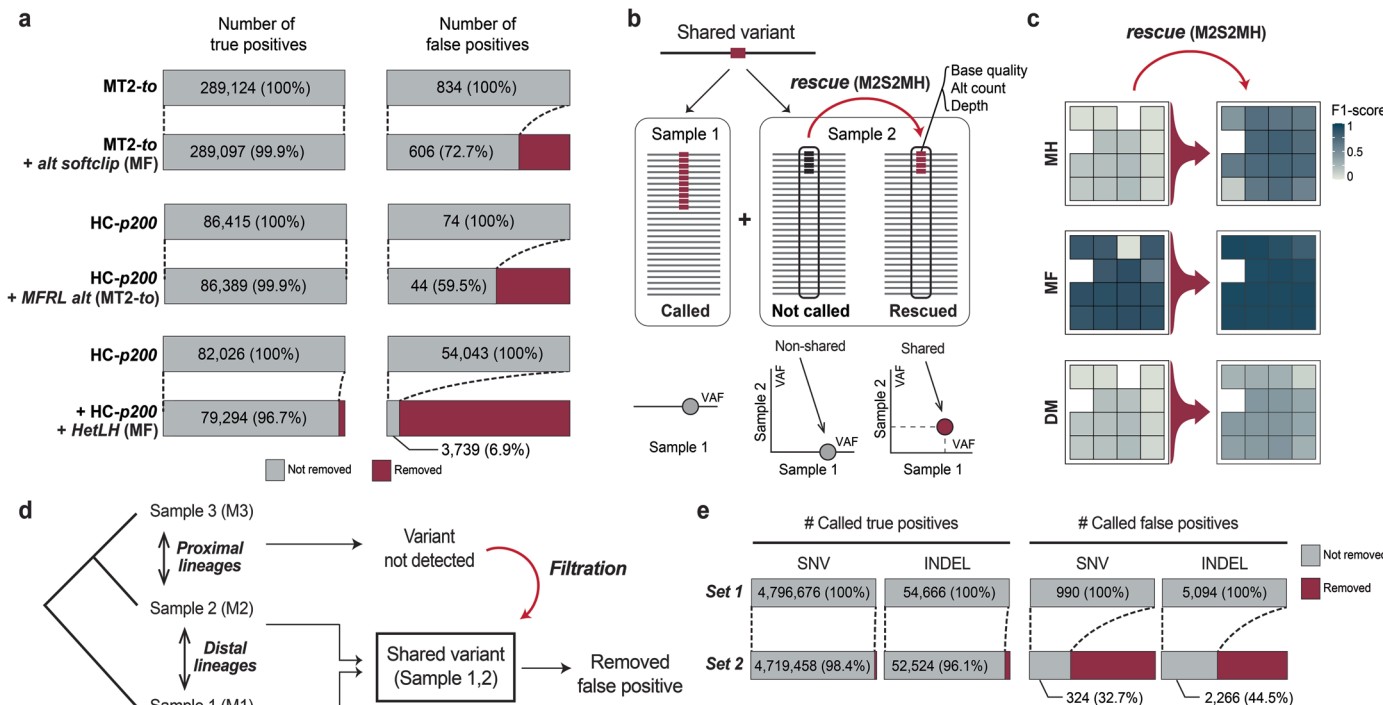

**Extended Data Fig. 4 | Additional strategies for mosaic variant calling.**
(**a**) Accuracy enhancement by the call set- and feature-level recombination of
multiple approaches. Three examples with the '*alt softclip*' in MF, '*MFRL alt*' in
MT2-*to*, and '*HetLH*' in MF, applied to the call sets of MT2-*to* with false positives
from non-variant sites, HC-*p200* with false positives from non-variant sites, and
HC-*p200* with germline false positives, respectively. Removed percentage of true
or false positives with the applications are shown in red. (**b**) A schematic overview
of the '*rescue*' procedure of M2S2MH approach. A shared variant that detected by
only in the either of two samples could be rescued with read-level information.
(**c**) The improvement in F1 scores of before and after applying the '*rescue*'
procedure to MH, MF, and DM, F1-scores shown in the identically partitioned
variant allele frequency (VAF) areas as in Fig. 3f. (**d**) Strategy for precision
enhancement of shared variant detection by utilizing multiple samples (≥3)
is shown. The distance in the developmental lineage of each sample and their
proximity in lineages are displayed. (**e**) Proportion of the removed true and false
positives when the new strategy (shown in **d**) was applied to shared SNVs and
INDELs. In total of 1,944 combinations generated by 39 reference standards in
three categories (9 M1, 12 M2, 18 M3) were tested with MT2-*to* call sets. All results
were generated with 1,100× data.

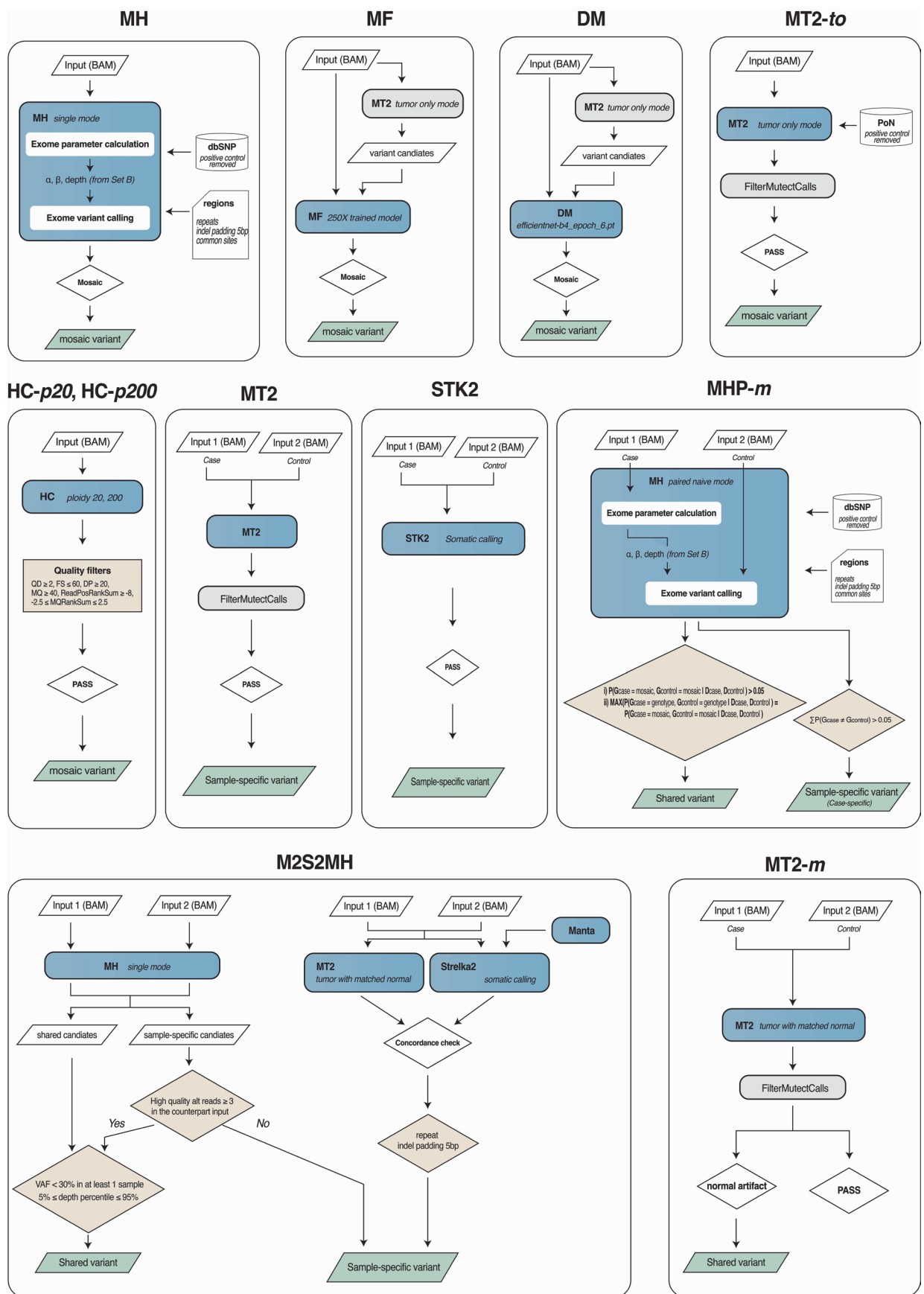

**Extended Data Fig. 5 | Schematic pipelines of the evaluated detection approaches.** Pipelines of eleven mosaic variant detection strategies utilized for benchmark are shown.

# Reporting Summary

## Statistics

For all statistical analyses, confirm that the following items are present in the figure legend, table legend, main text, or Methods section.

| n/a | Confirmed | |
|---|---|---|
| ☐ | ☒ | The exact sample size (*n*) for each experimental group/condition, given as a discrete number and unit of measurement |
| ☐ | ☒ | A statement on whether measurements were taken from distinct samples or whether the same sample was measured repeatedly |
| ☐ | ☒ | The statistical test(s) used AND whether they are one- or two-sided<br>*Only common tests should be described solely by name; describe more complex techniques in the Methods section.* |
| ☐ | ☒ | A description of all covariates tested |
| ☐ | ☒ | A description of any assumptions or corrections, such as tests of normality and adjustment for multiple comparisons |
| ☐ | ☒ | A full description of the statistical parameters including central tendency (e.g. means) or other basic estimates (e.g. regression coefficient) AND variation (e.g. standard deviation) or associated estimates of uncertainty (e.g. confidence intervals) |
| ☐ | ☒ | For null hypothesis testing, the test statistic (e.g. $F$, $t$, $r$) with confidence intervals, effect sizes, degrees of freedom and $P$ value noted<br>*Give P values as exact values whenever suitable.* |
| ☐ | ☒ | For Bayesian analysis, information on the choice of priors and Markov chain Monte Carlo settings |
| ☐ | ☒ | For hierarchical and complex designs, identification of the appropriate level for tests and full reporting of outcomes |
| ☐ | ☒ | Estimates of effect sizes (e.g. Cohen's *d*, Pearson's *r*), indicating how they were calculated |

*Our web collection on statistics for biologists contains articles on many of the points above.*

## Software and code

Policy information about availability of computer code

| Data collection | No software used |
|---|---|
| Data analysis | MosaicHunter (v.1.0), MosaicForecast (v.0.0.1), DeepMosaic (v.0.0), Mutect2 (4.1.9.0), GATK HaplotypeCaller (4.1.8.0), Strelka2 (v.2.9.10), Manta (v.1.6.0), and nVennR (0.2.3).<br>The codes used for analyses are provided in GitHub (https://github.com/hiyoothere/Benchmarking-mosaic-variant-detection). |

For manuscripts utilizing custom algorithms or software that are central to the research but not yet described in published literature, software must be made available to editors and reviewers. We strongly encourage code deposition in a community repository (e.g. GitHub). See the Nature Portfolio guidelines for submitting code & software for further information.

## Data

Policy information about availability of data

All manuscripts must include a data availability statement. This statement should provide the following information, where applicable:
- Accession codes, unique identifiers, or web links for publicly available datasets
- A description of any restrictions on data availability
- For clinical datasets or third party data, please ensure that the statement adheres to our policy

The source data was acquired from Sequence Read Archive under the accession code [PRJNA758606]

# Human research participants

Policy information about studies involving human research participants and Sex and Gender in Research.

| | |
|---|---|
| Reporting on sex and gender | N/A |
| Population characteristics | N/A |
| Recruitment | N/A |
| Ethics oversight | N/A |

Note that full information on the approval of the study protocol must also be provided in the manuscript.

# Field-specific reporting

Please select the one below that is the best fit for your research. If you are not sure, read the appropriate sections before making your selection.

☒ Life sciences        ☐ Behavioural & social sciences        ☐ Ecological, evolutionary & environmental sciences

For a reference copy of the document with all sections, see nature.com/documents/nr-reporting-summary-flat.pdf

# Life sciences study design

All studies must disclose on these points even when the disclosure is negative.

| | |
|---|---|
| Sample size | N/A |
| Data exclusions | N/A |
| Replication | N/A |
| Randomization | N/A |
| Blinding | N/A |

# Reporting for specific materials, systems and methods

We require information from authors about some types of materials, experimental systems and methods used in many studies. Here, indicate whether each material, system or method listed is relevant to your study. If you are not sure if a list item applies to your research, read the appropriate section before selecting a response.

## Materials & experimental systems

| n/a | Involved in the study |
|---|---|
| ☒ ☐ | Antibodies |
| ☒ ☐ | Eukaryotic cell lines |
| ☒ ☐ | Palaeontology and archaeology |
| ☒ ☐ | Animals and other organisms |
| ☒ ☐ | Clinical data |
| ☒ ☐ | Dual use research of concern |

## Methods

| n/a | Involved in the study |
|---|---|
| ☒ ☐ | ChIP-seq |
| ☒ ☐ | Flow cytometry |
| ☒ ☐ | MRI-based neuroimaging |

