## [Peer Review File · Nature Methods]

Peer Review Information

Manuscript Title: Comprehensive benchmarking and guidelines of mosaic variant calling strategies

Corresponding author name(s): Sangwoo Kim

Editorial Notes:

Reviewer Comments & Decisions:

Decision Letter, initial version:
--

5th Feb 2023

Dear Professor Kim,

Your Analysis, "Robust benchmark to evaluate and improve mosaic variant calling strategies", has now been seen by 3 reviewers. As you will see from their comments below, although the reviewers find your work of considerable potential interest, they have raised a number of concerns. We are interested in the possibility of publishing your paper in Nature Methods, but would like to consider your response to these concerns before we reach a final decision on publication.

We therefore invite you to revise your manuscript to fully address all these concerns with additional analysis and other revisions.

* include a point-by-point response to the reviewers and to any editorial suggestions

* please underline/highlight any additions to the text or areas with other significant changes to facilitate review of the revised manuscript

- * address the points listed described below to conform to our open science requirements
- * ensure it complies with our general format requirements as set out in our guide to authors at www.nature.com/naturemethods
- * resubmit all the necessary files electronically by using the link below to access your home page

[REDACTED]

We hope to receive your revised paper within eight weeks. If you cannot send it within this time, please let us know. In this event, we will still be happy to reconsider your paper at a later date so long as nothing similar has been accepted for publication at Nature Methods or published elsewhere.

OPEN SCIENCE REQUIREMENTS

REPORTING SUMMARY AND EDITORIAL POLICY CHECKLISTS

DATA AVAILABILITY

All novel DNA and RNA sequencing data, protein sequences, genetic polymorphisms, linked genotype and phenotype data, gene expression data, macromolecular structures, and proteomics data must be deposited in a publicly accessible database, and accession codes and associated hyperlinks must be provided in the "Data Availability" section.

CODE AVAILABILITY

Please include a "Code Availability" subsection in the Online Methods which details how your custom code is made available. Only in rare cases (where code is not central to the main conclusions of the paper) is the statement "available upon request" allowed (and reasons should be specified).

For more information on our code sharing policy and requirements, please see: <https://www.nature.com/nature-research/editorial-policies/reporting-standards#availability-of-computer-code>

MATERIALS AVAILABILITY

ORCID

Sincerely,

Lin Tang, PhD
Senior Editor
Nature Methods

Reviewers' Comments:

Reviewer #1:

Remarks to the Author:

In the submitted manuscript, Ha and colleagues describe a study on using simulated somatic mutations to benchmark existing callers and propose improvements. Studying somatic mosaicism has gained attention in recent years, and such studies are needed to further advance the field. The simulated set of somatic mutations is described in a previous publication (ref. 18). The set is constructed with 39 distinct mixtures of DNA for well-characterized cell lines and considers unique germline variants in each of the mixed lines to be somatic mutations. The fraction of a cell line in a mixture determines the frequency of simulated somatic mutations in the mixture. High number of mixtures was produced to simulate somatic mutations at various frequencies and to simulate relatedness between the mixtures through sharing the same simulated somatic mutations. Cell lines were characterized by deep (i.e., 1100× depth) exome sequencing, and exomes of the mixtures produced were also sequenced to that depth. Using the simulated set of somatic mutations (i.e., SNVs and indels), the set of non-mutated sites, and the set of germline variants, the authors evaluated the performance of existing methods used to discover somatic mutations. In the manuscript, they also

discuss which parameters of mutation calls can be leveraged to improve the calling of somatic mutations as well as new strategies for reducing false-positive calls. The manuscript is not only timely but also written well and presents novel ideas and analyses. Altogether, despite some limitations and both explicit and implicit biases, some of which I discuss below, the manuscript promises to be a great addition to the literature and will be of interest to researchers who study somatic mosaicism and cancer.

Comments on how to improve the manuscript

* How the benchmark set was derived is briefly described, and although an extended description is not necessary, the mixture categories are not defined or described, which negatively impacts comprehension. Only after having carefully read Reference 18, I can understand what M1, M2, and M3 are. Perhaps these mixture categories can be described in Figure 1b.

* The stated number of simulated somatic mutations (i.e., 374,565) is confusing, particularly in comparison with the number of non-variant (i.e., 35,113,417) and germline (i.e., 19,936) sites. According to Reference 18, the number of actual sites with simulated somatic mutations is approximately 20,000, but each site simulates mutations in approximately 20 mixtures. Perhaps this should be better explained, together with a description of the mixtures (see my previous comment).

* Cell lines used to prepare mixtures and the mixtures themselves were exome-sequenced. However, because capturing targeted DNA regions is known to involve various biases—for instance, indels are not captured well, and their frequency is systematically underestimated—I am concerned that such a characterization may have had inexplicit biases on the analyses in the study and on ideas presented in the manuscript. I think that the concern warrants discussion in the manuscript.

* Precision and F1 score depend on the number of true positives. Because the total number of simulated mosaic mutations is very high (374,565), I am also concerned about how to treat numbers reported for those two measures. Only in the Methods do the authors describe that they adjusted true positives for the expected number of somatic mutations. Although doing so is acceptable, I think that the adjustment should also be mentioned before describing the results of benchmarking. Beyond that, the issue should be discussed in Discussion as well, for I do not think that the field has knowledge of how to properly estimate the expected number of somatic mutations. In my view, precision and F1 score should be interpreted as relative values.

* In relation to my previous comment, it seems that the weights for adjusting true positives not 252.75 and 81.63 but $1/252.75$ and $1/81.63$ instead. Or am I mistaken?

* The strategy for using sample relatedness to reduce false positives is described poorly. In particular, it is not clear what the distal or proximal lineage/samples are. Only after understanding the meaning of mixing categories M1, M2, and M3 (see my first comment) could I guess what the authors mean by “distal lineage” and “proximal lineage.” Even so, how do the authors formally define those two terms?

* The manuscript should declare the data’s availability.

Other minor comments:

* Figure 2f is unclear, and I have no idea how to improve it.

* Lines 154–155: It is not apparent to me that “three joint-genotyping methods (MT2, STK2, and

M2S2MH) 155 marked the best accuracy (Fig. 3c and 3d)."

* Line 158: What do you mean by "supplementary figure X"?

Reviewer #2:

Remarks to the Author:

The authors present a benchmarking experiment for multiple mosaic variant callers against their previously published benchmark from mixtures of germline variants in normal samples. This is an interesting approach to benchmarking mosaic variant detection, similar to what has been used for somatic variants previously, and I think it could be quite valuable. However, it has some major limitations that should be mitigated and/or highlighted much more clearly. In particular, I suspect that the use of germline variants could bias the results for or against some of the methods, as detailed below. There are also a number of other improvements outlined below:

1. "the overall classification power measured in area under the precision-recall curve (AUPRC) was the best in MF and HC-p200 (Fig. 2c), indicating a margin for improvement by parameter optimization." This statement incorrectly refers to Fig 2c instead of 2b, and it is also not a correct statement for the SNV AUPRC except >25% VAF. For all the other VAFs, MT2-to appears to be the best for AUPRC.
2. "For example, the F1-score of MF was increased by additional 0.18, 0.34, and 0.42, at 250x, 500x, and 1,100x, respectively, compared to 0.20 at 125x." This is a confusing way to state this - maybe give the actual F1 scores for each like is done in the next sentence, instead of the "additional" amount?
3. "Generating in silico simulated datasets (e.g., mixing BAM files) is also a simple but powerful method¹⁶, however, we noted that the error profiles, especially the sequencing errors, are restricted to the source data and cannot represent the real-world level artifacts (Supplementary Fig. 9). We believe that our approach to constructing a mixture-derived standard reference is the most appropriate way for securing both the scale and robustness for mosaic variant analysis." I am not sure what the authors mean by in silico vs mixture-derived standards, or what Supp Fig 9 represents. Could the authors explain more the difference between these approaches and exactly how they did the in silico experiment, since this isn't described in the methods?
4. "Finally, the composition of the standard reference can affect the results, such as the distribution of VAFs in the datasets, sequencing platform, read length, and error profiles. For example, the cumulative cell line mixing (Fig. 1) produced a large set of robust mosaic variants, but it could also alter the composition of germline variant sites. We resolved the issue of loss of pure germline sites by preparing a separate set of reference standards that recovered germline sites¹⁸. Although the effect is assumed as ignorable (see Supplementary Notes), we should be aware of it when evaluating tools that consider the alignment patterns of the flanking regions." In the Supp Note, it would be a stronger argument if the authors would show the distribution of distances of each positive control to the nearest germline variant.
5. "The population frequency data was needed for default settings, dbSNP(b154) for Mosaic Hunter and panel of normal (1000g_pon.hg38.vcf.gz) from GATK resource bundle (<https://console.cloud.google.com/storage/browser/genomics-public-data/resources/broad/hg38/v0>) for Mutect2 were applied after the removal of positive controls". This should be highlighted in the manuscript as a significant drawback of the benchmarking approach used in this manuscript. This could introduce biases in the analysis. For example, excluding the positive controls might enhance performance for the benchmark relative to true performance if some real mosaic variants match variance in dbSNP or 1000 Genomes.

6. Even more importantly, DeepMosaic used population information in the training so I suspect its poor performance might be caused by the design of this benchmark, which actually uses germline variants but mimics mosaic variants by mixing. How do the authors know that DeepMosaic's performance is not hindered by using germline variants as positive controls?
7. Does MosaicForest use population information?
8. "Segmental duplication and simple repeats from UCSC were used whenever removing repeats to filter out confounding regions was recommended." For which methods was this done, and were these regions also removed from the benchmark, or were these counted as FNs?
9. The only mention of code availability was the link to github ref 31 "Yoo-Jin Ha, J.K., Seungseok Kang, Junhan Kim, Se-Young Jo, and Sangwoo Kim Benchmark for improved accuracy in mosaic variant calling. GitHub (2021)." I could not find this github site when searching for it.
10. In this approach, my understanding is that there are many more mosaic variants than there would be in a real sample, which would result in substantially higher estimates of precision than in a real sample. Instead of precision, maybe a better metric would be FPs/Mbp or even just # of FPs?
11. Were any of the callers used to form the benchmark also evaluated in this work?
12. How conservative is the benchmark? For example, does it mostly exclude repetitive regions like segmental duplications or other regions prone to systematic errors like homopolymers and tandem repeats?
13. It should be clear in the abstract that the authors are testing exome sequencing data.
14. It's not clear to me how coverage variability in exome sequencing affects these results. Do the authors exclude variants below a certain coverage or deal with edges of captured regions in a different way?
15. When using a benchmark to evaluate methods it is important to ensure it is reliably identifying FPs and FNs. Could the authors manually curate a subset of the FPs and FNs from each callset in a genome browser like IGV to ensure these are actual errors in the callset and not errors in the benchmark or how the authors ran the tools or performed comparisons?
16. In the title, I think it would be more accurate to say "Extensive benchmarking (or evaluation)" rather than "Robust benchmark" because the focus of this manuscript is on the performance evaluation rather than making a benchmark, and it is not clear how robust the approach is given its use of germline variants.
17. It appears that 3 of the 5 cell lines used in this work don't have public exome or genome sequencing data apart from the authors' work. One appears to only have restricted access for exome data, so others have determined that consent is not appropriate for public access (<https://www.ncbi.nlm.nih.gov/sra/?term=HBEC3-KT>). Informed consent is particularly important if these are to be established as widely-used benchmarks. Could the authors clarify what human subjects review that was done to determine that these data could be made public, and whether the individuals from whom these cell lines were derived were consented for public release of genomic data?

Reviewer #3:

Remarks to the Author:

A. Summary of the key results

In this manuscript, Ha et al. performed a detailed benchmark of 11 variant calling strategies based on a benchmark dataset the same group generated previously (PMID 35115554) and provided their best practice and condition-dependent strength and weakness for users. One of the major conclusions, which is also commonly accepted in the field, is that different algorithm finds distinct parts of the true

mosaic variants and can be complemented by others with different strategies. The authors also provided recommended best practices based on their benchmark dataset, which is very useful for the field, if their benchmark dataset is reflecting real-world data features.

B. Originality and significance: if not novel, please include reference

Calling postzygotic mosaic mutations (from non-cancer tissues) is very challenging, and is very important in this field. A comprehensive benchmark for different mosaic mutation calling strategies is very important not only for the mutation field, but also for researchers studying developmental biology, clinical genetics, and beyond. The originality of the study is valid as no benchmark for non-cancer mosaic mutations is carried out for different strategies other than tool builders themselves.

C. Data & methodology: validity of approach, quality of data, quality of presentation

While the data looks very informative based on their benchmark dataset, and calibrations are carefully carried out, I have some concerns:

a. The benchmark dataset is generated by mixing 1100x WGS from cell lines with different haplotypes, will this affect the performance of any of the strategies? The features provided in Figure 4, as well as lines 209-213 also bring up a concern that haplotype-based features might bias this assessment. Do the authors have an independent biological dataset other than the simulated dataset to validate their conclusions? Also, Supplementary Figure 1 showed distributions of VAFs for true positive and true negatives, are the distributions of these VAFs resembling VAFs from the real data? Are VAFs the only features of the benchmark dataset that is similar to existing data, such as published human exomes sequenced in developing tissues or cancer?

b. The author provided a thorough comparison of different strategies, do they have a reason for not including CaVEMan (PMID 34433962 and 34433962) which is used in a good amount of high-impact publications?

c. The author showed convincingly that higher sequencing depth is in general increasing the performance of mosaic variant detection. With the dropping of sequencing price it will be more feasible to sequence all the samples at a much higher depth, most of the currently existing exome and genome data, however, are way below the minimum depth tested in this manuscript (125X). For exomes, most are below 100x, for clinical diagnostic exomes that are massively generated they are normally below 60x. Do the authors not recommend using any of the methods in the existing data or recommend to re-sequence the existing samples for mosaic detection? Do they have some estimates on existing low-depth samples? Also, for most of the parts in the manuscript, the authors didn't specify that they are working on ultra-deep exome data. Do the authors expect similar performance for genomes or capture panels?

d. Line 85: citation 8 used poldy 50, does the author use poldy 200 to compensate the 1100x WES to get to lower AF, or do they have evidence that p200 performs better than p50?

e. Line 170-172, is this performance different because MF is running based on M2 results? If MF is running based on HC-p20 or HC-p200 will it improve the performance?

f. Line 373: "PASS" recommended in DM tutorial.

D. Appropriate use of statistics and treatment of uncertainties

Most of the statistics are well performed in the current manuscript, but there is some inconsistency between the currently used data and the published data from the author, claiming to be the same dataset.

a. The ground truth set is a unique resource and the basis for this study, however, in the original publication (PMID 35115554) where the author built their reference standard, there are 386,613 positives and 35,113,417 non-variants, but in the current manuscript there are 390,153 positives and

35,208, 888 negative positives, in lines 61-63, the authors might want to explain what are the differences between the two datasets that are described?

E. Conclusions: robustness, validity, reliability

a. The benchmark analyses and the recommendations in box 1 are highly appreciated, from a user perspective, it would be very important to know whether the same strategy applies to real-world data, published by the authors (PMID 36121845) for example.

F. Suggested improvements: experiments, data for possible revision

a. Comparison with CaVEMan as an important independent strategy

b. Inclusion of experimental data (PMID 36121845 for example) to validate the major conclusions or additional data to support that the benchmark dataset is providing enough features that resemble real data.

G. References: appropriate credit to previous work?

a. This study focuses on SNV/INDELS, the author might want to point this out so that readers can get better use of their benchmark, and also maybe specify do these strategies also work on non-human organisms (PMID: 35418684).

H. Clarity and context: lucidity of abstract/summary, appropriateness of abstract, introduction and conclusions

Overall the clarity is good, I have some minor comments:

a. Line 17: I think mosaic variants can exist in germline and somatic tissues, so this sentence is not very accurate. Heterozygous, homozygous, and mosaic variants are more comparable concepts?

b. Line 18: The focus of different studies is affecting the use of methods, not sure chaotic is a good word for the field

c. Figure 3: What's the depth shown here? Also in the legend: log₁₀ scale

Author Rebuttal to Initial comments

Reviewers' Comments:

Reviewer #1:

Remarks to the Author:

In the submitted manuscript, Ha and colleagues describe a study on using simulated somatic mutations to benchmark existing callers and propose improvements. Studying somatic mosaicism has gained attention in recent years, and such studies are needed to further advance the field. The simulated set of somatic mutations is described in a previous publication (ref. 18). The set is constructed with 39 distinct mixtures of DNA for well-characterized cell lines and considers unique germline variants in each of the mixed lines to be somatic mutations. The fraction of a cell line in a mixture determines the frequency of simulated somatic mutations in the mixture. High number of mixtures was produced to simulate somatic mutations at various frequencies and to simulate relatedness between the mixtures through sharing the same simulated somatic mutations. Cell lines were characterized by deep (i.e., 1100× depth) exome sequencing,

and exomes of the mixtures produced were also sequenced to that depth. Using the simulated set of somatic mutations (i.e., SNVs and indels), the set of non-mutated sites, and the set of germline variants, the authors evaluated the performance of existing methods used to discover somatic mutations. In the manuscript, they also discuss which parameters of mutation calls can be leveraged to improve the calling of somatic mutations as well as new strategies for reducing false-positive calls. The manuscript is not only timely but also written well and presents novel ideas and analyses. Altogether, despite some limitations and both explicit and implicit biases, some of which I discuss below, the manuscript promises to be a great addition to the literature and will be of interest to researchers who study somatic mosaicism and cancer.

Answer to the General comment:

We greatly appreciate the positive evaluation of our study. We understood the limitations that the reviewer pointed out, and fully addressed in this revision, which made the manuscript much clearer. We hope that the revised manuscript is publishable and look forward to sharing our study with researchers in the relevant fields.

Comments on how to improve the manuscript

1. How the benchmark set was derived is briefly described, and although an extended description is not necessary, the mixture categories are not defined or described, which negatively impacts comprehension. Only after having carefully read Reference 18, I can understand what M1, M2, and M3 are. Perhaps these mixture categories can be described in Figure 1b.

Answer to Q1 – We do agree with the reviewer’s comment. In this revision, we made the overall benchmark preparation procedure including the mixture categories better understandable without referring to the Ref 18. We substantially revised **Fig 1b** to include the definition and the final form of the mixtures (M1, M2, and M3) and how they were generated from the normal cell lines. The total number of mutations with the variant allele frequencies (VAF) were also added. Accordingly, the beginning part of the Introduction has been completely rewritten (line 59-76). We thank the reviewer for the advice.

2. The stated number of the simulated somatic mutations (i.e., 374,565) is confusing, particularly in comparison with the number of non-variant (i.e., 35,113,417) and germline (i.e., 19,936) sites. According to Reference 18, the number of actual sites with simulated somatic mutations is

approximately 20,000, but each site simulates mutations in approximately 20 mixtures. Perhaps this should be better explained, together with a description of the mixtures (see my previous comment).

Answer to Q2 – We apologize for the unclear description about the numbers. The reviewer is right that the number of actual sites with simulated is approximately 20,000, and the total number of mutations is calculated by multiplying the number of samples (=39); we think that this is a valid description because all the mutations are independently generated and have different VAFs in each sample, in contrast to the non-variant and germline variants. To explain this better, we wrote both numbers (the total number of mutation and the unique sites) side-by-side in the revised **Figure 1b** and the main text (line 66-76).

In addition, in this revision, we change the number of simulated somatic mutation to 345,552 that were actually used in this benchmark, which is a high confidence subset of 374,565. We also clarified this in the revised manuscript (line 71-73 and line 401-403 in **Methods**). We thank the reviewer for the suggestions.

3. Cell lines used to prepare mixtures and the mixtures themselves were exome-sequenced. However, because capturing targeted DNA regions is known to involve various biases—for instance, indels are not captured well, and their frequency is systematically underestimated—I am concerned that such a characterization may have had inexplicit biases on the analyses in the study and on ideas presented in the manuscript. I think that the concern warrants discussion in the manuscript.

Answer to Q3 – We fully agree with the reviewer that capture efficiency in exome sequencing is an implicit external factor that leads to various biases. To identify the effect more precisely, we newly added an analysis on the read depth distribution in SNVs and INELS (please see the below Figure) and confirmed the (1) variance towards genomic positions and (2) consistent lower depth in INDEL sites. Indeed, the lowered read depth would have negatively affected the overall accuracy. Because this effect is an intrinsic characteristic of capture-based whole exome sequencing that reflects the real-world performance, we believe that no further recalibration is required in this benchmark. However, as the reviewer pointed out, we discussed this concern in Discussion (line 317-322) with the analysis result in **Supplementary Notes** (section 5). We appreciate the reviewer’s valuable comment.

Depth at positive controls	SNV	INDEL
1st quantile	358.0	121.0
2nd quantile (Median)	560.0	139.0
3rd quantile	805.0	684.0
Mean	616.2	139.0

4. Precision and F1 score depend on the number of true positives. Because the total number of simulated mosaic mutations is very high (374,565), I am also concerned about how to treat numbers reported for those two measures. Only in the Methods do the authors describe that they adjusted true positives for the expected number of somatic mutations. Although doing so is acceptable, I think that the adjustment should also be mentioned before describing the results of benchmarking. Beyond that, the issue should be discussed in Discussion as well, for I do not think that the field has knowledge of how to properly estimate the expected number of somatic mutations. In my view, precision and F1 score should be interpreted as relative values.

Answer to Q4 – We completely agree with the reviewer that the precision and F1-score is dependent on the ratio between control positives and negatives, and should be interpreted as a relative measure. As suggested by the reviewer, we briefly mentioned the recalibration process before the benchmark results (line 73-75), and also add more detailed description in Discussion (line 343-345).

In addition, we newly introduced a new measure (FP/Mbp: number of false positives per Mb), suggested by another reviewer. The FP/Mbp is in principal independent from the number of true positives in the dataset and illustrates an absolute false positivity of the tools. All measured FPs/Mbp values were newly added in **Figure 2** and **Supplementary Figure 2, 6, and 7** for all

sequencing depths. We expect that the addition of the new measure would complement the relativity of the conventional statistics. We thank the reviewer for the valuable comment.

5. In relation to my previous comment, it seems that the weights for adjusting true positives not 252.75 and 81.63 but 1/252.75 and 1/81.63 instead. Or am I mistaken?

Answer to Q5 – We thank the reviewer’s comment. The reviewer is right, and we corrected the numbers.

6. The strategy for using sample relatedness to reduce false positives is described poorly. In particular, it is not clear what the distal or proximal lineage/samples are. Only after understanding the meaning of mixing categories M1, M2, and M3 (see my first comment) could I guess what the authors mean by “distal lineage” and “proximal lineage.” Even so, how do the authors formally define those two terms?

Answer to Q6 – We apologize for the unclear description about the sample relatedness and distance. In this revision, we formally defined the two terms like below. First, we define the distance between a sample pair $p_1 = (A, B)$ in a phylogenetic tree using their most recent common ancestor $c(A, B)$ (please see the Figure below). We say a sample pair $p_1 = (A, B)$ is relatively proximal to a sample pair $p_2 = (A, C)$, if $c(p_2) = c(p_1, p_2)$ and $c(p_1) \neq c(p_1, p_2)$. In other words, a sample pair is in a more proximal lineage if they are branched out more recently. For example, two samples developed from same germ layer (e.g., brain and skin; both from ectoderm) are more proximal than ones from two different germ layers (e.g., brain and skeletal muscle; ectoderm and mesoderm). Likewise, the relative distance of the samples in three mosaic types (M1-M3) could be defined, wherein samples in a more proximal lineage indeed had more shared variants. Although the cellular heterogeneity in tissue (e.g., blood in tissue) can limit the clarification of the origin of each tissue in real-world, using a subset of well-characterized somatic mutations could be utilized for reconstructing the developmental lineages in recent studies. We revised our manuscript in line 260-263 and added Supplementary Notes (section 4) to include the formal definition and the detailed description. We again thank the reviewer for the question.

7. The manuscript should declare the data's availability.

Answer to Q7 – We apologize for the missing information about the data availability. We added the data availability information with the data repository information and all the computational codes that were used in this study (manuscript page 22).

Other minor comments:

* Figure 2f is unclear, and I have no idea how to improve it.

Answer to minor Q1 – We newly added explanatory lines to Figure 2f to clarify that each colored part depicts depth-specific true and false positive calls. Detailed descriptions are also added in the manuscript line 135-138 and the figure legend.

* Lines 154–155: It is not apparent to me that “three joint-genotyping methods (MT2, STK2, and M2S2MH) 155 marked the best accuracy (Fig. 3c and 3d).”

Answer to minor Q2 – Thanks to the comment, we revised our manuscript to clarify that the three of the joint-genotyping methods showed superior accuracy over others, benefitted by considering samples together in calculation (manuscript line 158 and 164-166)

* Line 158: What do you mean by “supplementary figure X”?

Answer to minor Q3 – Sorry for the inconvenience. We revised the X as 6 in the manuscript.

Reviewer #2:

Remarks to the Author:

The authors present a benchmarking experiment for multiple mosaic variant callers against their previously published benchmark from mixtures of germline variants in normal samples. This is an interesting approach to benchmarking mosaic variant detection, similar to what has been used for somatic variants previously, and I think it could be quite valuable. However, it has some major limitations that should be mitigated and/or highlighted much more clearly. In particular, I suspect that the use of germline variants could bias the results for or against some of the methods, as detailed below. There are also a number of other improvements outlined below:

Answer to the General comment:

We greatly appreciate the overall positive evaluation of our study. We do agree with the reviewer that this first benchmark on mosaic variant analysis would be highly useful to researchers in the genomic field. And we also noticed and understood all the concerns and limitations that the reviewers pointed out, most of which are important questions and could be fully addressed in this revision. For a few concerns, such as biases from using germline-derived ground truth variants, we could clarify and/or prove the validity of our approach with additional descriptions and analyses. Overall, all the comments from this reviewer were insightful and greatly improved the manuscript. We hope that the revised manuscript is publishable and look forward to sharing our study with researchers in the relevant fields.

1. “the overall classification power measured in area under the precision-recall curve (AUPRC) was the best in MF and HC-p200 (Fig. 2c), indicating a margin for improvement by parameter optimization.” This statement incorrectly refers to Fig 2c instead of 2b, and it is also not a correct statement for the SNV AUPRC except >25% VAF. For all the other VAFs, MT2-to appears to be the best for AUPRC.

Answer to Q1 – We appreciate the reviewer’s correction and clarified that the AUPRC of MF and HC-p200 performed the best at only high-VAF ($\geq 25\%$) mosaic SNV calling (manuscript line 108-110).

2. “For example, the F1-score of MF was increased by additional 0.18, 0.34, and 0.42, at 250 \times , 500 \times , and 1,100 \times , respectively, compared to 0.20 at 125 \times .” This is a confusing way to state this - maybe give the actual F1 scores for each like is done in the next sentence, instead of the “additional” amount?

Answer to Q2 – We agree with the reviewer’s comment that the gain of F1-score for each sequencing depth can confuse readers. We added the actual F1-score in each depth in the manuscript (line 131-133) with the relative gain from the lowest depth (125×). We thank the reviewer for the suggestion.

3. “Generating *in silico* simulated datasets (e.g., mixing BAM files) is also a simple but powerful method¹⁶, however, we noted that the error profiles, especially the sequencing errors, are restricted to the source data and cannot represent the real-world level artifacts (Supplementary Fig. 9). We believe that our approach to constructing a mixture-derived standard reference is the most appropriate way for securing both the scale and robustness for mosaic variant analysis.” I am not sure what the authors mean by *in silico* vs mixture-derived standards, or what Supp Fig 9 represents. Could the authors explain more the difference between these approaches and exactly how they did the *in silico* experiment, since this isn’t described in the methods?

Answer to Q3 – We apologize for the unclear description about the comparison between *in silico*-generated and mixture-derived reference standards. Here, we fully describe the difference and the description about the Supplementary Fig 9 (Supplementary Fig. 14 in revised manuscript), and how we reflected them in the revised manuscript.

The *in silico* mixture dataset is another version of the ground truth variant set that was generated by mixing BAM files (instead of DNA) of the six cell lines. Like in the true reference standard, we generated 39 *in silico* mixtures with the same designated proportions. This would be a much simpler and low-cost procedure than the mixing DNAs, because we do not conduct actual sequencing of the 39 samples. Also, it is much more convenient to achieve accurate target mixture-proportion. This is why we compared the *in silico* mixture set with the real DNA-mixtures.

When we assessed the error profiles of the *in silico* mixtures, however, we found that the error profiles (e.g., the positions and types of the sequencing errors) are confined to the ones in the original cell line sequencing, which greatly limit the variety of the noises (Supplementary Fig 14 right). Consequently, the overall evaluation cannot be conducted with sufficient noise sources and levels that we face in the real-world settings. Therefore, we concluded that the mixture-derived reference standard is a more appropriate approach for a benchmark study.

We added the detailed procedures for generating the *in silico* mixture dataset in the Supplementary Notes (section 3), and revised the main text (line 292-294) to more properly explain the goal and the conclusion of the analysis. We appreciate the valuable comment from the reviewer.

4. “Finally, the composition of the standard reference can affect the results, such as the distribution of VAFs in the datasets, sequencing platform, read length, and error profiles. For example, the cumulative cell line mixing (Fig. 1) produced a large set of robust mosaic variants, but it could also alter the composition of germline variant sites. We resolved the issue of loss of pure germline sites by preparing a separate set of reference standards that recovered germline sites¹⁸. Although the effect is assumed as ignorable (see Supplementary Notes), we should be aware of it when evaluating tools that consider the alignment patterns of the flanking regions.” In the Supp Note, it would be a stronger argument if the authors would show the distribution of distances of each positive control to the nearest germline variant.

Answer to Q4 – We thank the reviewer for the comment. As the reviewer suggested, we draw the distribution of distances from the true variants to the nearest germline variants (please refer to the below Figure). Despite the average distance was large (1,772 bp), we found that approximately 39% of the true variants were accompanied with germline variants within a read length (150 bp), unlike our previous assumption. We do not think it is an anomaly only present in the dataset, because the presence of germline or subclonal mutations within a read is routinely observed (which is also an essential source for haplotype phasing), but we acknowledged a potential over-representation that comes from multiple cell lines. Therefore, we conducted further analysis in this revision.

To measure the potential effect, we calculated sensitivities of all tools for only variants with proximal (<150 bp) germline variants. In all tools except MT2 and MF, no significant differences were observed. In contrast, sensitivity was approximately 4.9% and 4.1% lower in MT2 and MF, implying a potential negative effect on these tools; we assume that the “clustered event” filtration strategy of MT2 makes these tools more sensitive to the presence of nearby germline variants. Again, 9-44% (27% in overall) of subclonal variants were observed with nearby germline variants¹, and the same decrease in sensitivity would be reproduced in real-world data. So, we assume the true effect from our study design should be partial. Also, the overall conclusion and the final best practice have not been affected because MT2 and MF were already top ranked with significant gaps from the others. However, based on the new results, we retracted the word

“ignorable” and discussed the potential effect as a limitation in Discussion (line 332-336). We again thank the reviewer for the suggestion.

Tools	Sensitivity (all)	Sensitivity (germline variants within 150 bp)	Difference
MH	0.2001	0.1959	0.0042
MT2	0.8402	0.7909	0.0493
HC20	0.2322	0.2382	-0.0060
DM	0.1025	0.0982	0.0042
MF	0.7624	0.7211	0.0412
HC200	0.2510	0.2578	-0.0068

5. “The population frequency data was needed for default settings, dbSNP(b154) for Mosaic Hunter and panel of normal (1000g_pon.hg38.vcf.gz) from GATK resource bundle (<https://console.cloud.google.com/storage/browser/genomics-public-data/resources/broad/hg38/v0>) for Mutect2 were applied after the removal of positive controls”. This should be highlighted in the manuscript as a significant drawback of the benchmarking approach used in this manuscript. This could introduce biases in the analysis. For example, excluding the positive controls might enhance performance for the benchmark relative to true performance if some real mosaic variants match variance in dbSNP or 1000 Genomes.

Answer to Q5 – We appreciate the important comment from the reviewer and apologize for the insufficient description. We took this comment seriously and would like to answer clearly with additional data. In conclusion, we are convinced that the bias from the way we generated true variants (germline variant-derived) is present, but small enough to make the performance evaluation reliable. Hence, our benchmark results successfully reflect the true performance that are measured on sporadic mosaic variant set (which is ideal but impossible to construct). Here, we would like to clarify the procedure and assess the true risks on evaluating the benchmarking approaches.

First of all, what we excluded are the *entries* in the population allele frequency (pAF) databases that are fed to each tool (e.g., dbSNP or panel_of_normal), *not* the variants; we modified the pAF *databases* not to contain the true positive sites. The purpose of this was to prevent each tool from treating the true positive variants as known SNP positions. By doing this, we assumed that germline variant-derived true variants are equivalent to sporadically generated variants, regarding to measuring variant calling accuracy (hypothesis 1). The important basis for the

hypothesis 1 is that no tool embeds genomic coordinates of SNP sites, but solely depend on external databases as input.

To confirm the hypothesis 1, we conducted an additional analysis in this revision. Without excluding the entries in the population databases, we conducted the same benchmark analysis on a subset of the true variants that are not located in SNP sites; the source of this non-SNP true variants is the rare germline variants in the original cell lines. We built one subset from each database (“panel_of_normal”: 1000g_pon.hg38.vcf.gz by Broad Institute, gnomAD: “hg38_gnomad211_gnomad.txt, and dbSNP build 154). We found that result is almost same (please see the Figure below), except a few intervals with small data points, which proves that the germline-derived construction did not lead to a bias.

Second, we assessed if the restriction of the true variant positions to the non-SNP sites leads to another type of bias, which we think the reviewer indicated in the question. We agree with the reviewer that mosaic variants that occur in known SNP sites have chances to be filtered out. And this may lead to overestimated sensitivity as our true variant sets do not allow the filtration. However, we also assumed that the actual effect of the filtration is limited due to the rarity of the SNP sites of sufficiently high pAF (e.g., $pAF > 0.001$) compared to the genome size (hypothesis 2).

To assess the hypothesis 2, we retrieved the sizes of three pAF databases (“panel_of_normal”, gnomAD, and dbSNP) by their allele frequencies. (please see the table below). We found that most of the entries were rare SNPs ($pAF < 0.001$). In total, 98,481,895 dbSNP and 51,447,633 gnomAD entries had $pAF > 0.001$. Thereby, we could calculate the probability that a random mosaic variant hits these sites as 3.1% and 1.6% (numbers divided by 3.2B). In other words, even without allowing SNP regions, our benchmark dataset emulates $> \sim 97\%$ of real-world cases, which makes the overall result reliable. We further assume that the true effect is even smaller because (1) not all the SNPs > 0.001 is filtered out (MosaicForecast uses the information

as an optional post-filter, other tools use it as rather one of the supportive factors within classification models (DeepMosaic) or in calculation of mosaic probabilities (MosaicHunter), along with more directly evident features from sequencing data). Also, (2) the actual match is even rarer; tools consider the ref-alt allele pair for match, which reduce the actual match by $\sim 1/3$. For PoN, all the matches are filtered regardless of pAF. So, the matching probability is 0.08% (2,609,566 divided by 3.2B).

pAF range	# dbSNP entries (%)	# gnomAD entries (%)	# PoN entries
0-0.000001	9,964,158 (1.69%)	98,698,804 (12.14%)	2,609,566
0.00001-0.000001	165,550,421 (28.15%)	339,141,353 (41.71%)	
0.0001-0.00001	195,698,500 (33.28%)	264,212,535 (32.50%)	
0.001-0.0001	118,355,282 (20.13%)	59,552,322 (7.32%)	
0.001-0.01	44,086,369 (7.50%)	27,301,443 (3.36%)	
0.01-0.1	8,094,187 (1.38%)	14,265,332 (1.75%)	
0.1-1	46,301,339 (7.87%)	9,880,858 (1.22%)	
Total	588,050,256 (100%)	813,052,647 (100%)	2,609,566 (100%)
Total (pAF < 0.001)	98,481,895 (16.7%)	51,447,633 (6.32%)	-

Finally, we newly added three independent datasets in this revision, to confirm if our benchmark result is consistent with real-world data, suggested by another reviewer. Despite the small size of the datasets, we could confirm that our results and the overall best practice guideline is valid. We believe this analysis also alleviates the bias issues and the concerns from the germline variant-derived ground truth.

By these results, we believe that the germline-derived generation of benchmark dataset sufficiently well emulated the real sporadic mosaic variants for the use of performance evaluation.

As we agree with the reviewer that this concern should be discussed, we added all of the new data and the analysis in Supplementary Notes (section 7), and potential limitations in the Discussion (line 336-342). We again thank the reviewer for the important comments that improved our manuscript.

6. Even more importantly, DeepMosaic used population information in the training so I suspect its poor performance might be caused by the design of this benchmark, which actually uses

germline variants but mimics mosaic variants by mixing. How do the authors know that DeepMosaic's performance is not hindered by using germline variants as positive controls?

Answer to Q6 – We thank the reviewer for the question. As we wrote in the previous answer, no tools use genomic coordinates of SNP sites. So, without external pAF databases, germline-derived mosaic variants are in principle equivalent to sporadic variants. We confirmed this with the main author of DeepMosaic, Dr. Xiaoxu Yang through the GitHub page (<https://github.com/VirginiaXu/DeepMosaic/issues/12>). As we mentioned in the Q5, tests on the independent biological datasets mostly reproduced the similar performances of the tools. Therefore, we do not think a performance of a specific tool (including DeepMosaic) is falsely affected by our study design.

7. Does MosaicForest use population information?

Answer to Q7 – Yes, MosaicForecast uses population information, not in the core model but as a post-filter (MF recommends filtering variants with $pAF > 0.001$). We did not apply this post-filter because MosaicForecast recommended it as an optional filtration step, outside of its model. Nevertheless, there is no actual effect of the filter, like all other tools, because removed pAF information from the external dbSNP database, whose overall impact is minor as we explained in Q5.

8. “Segmental duplication and simple repeats from UCSC were used whenever removing repeats to filter out confounding regions was recommended.” For which methods was this done, and were these regions also removed from the benchmark, or were these counted as FNs?

Answer to Q8 – The repeat region filtration was formerly applied to generate the high confidence ground truth set (374,565 true positives). Therefore, any of these regions were counted as FNs for all tools. The statement that the reviewer referred was a part of official variant calling protocols that we used for benchmark, given by the tools; three of the mosaic callers (MosaicHunter, MosaicForecast, and DeepMosaic) include this filtration step inside their calling procedures. Again, the caller-specific filtration did not affect the number of FNs, because they were already removed in the ground truth preparation. We thank the reviewer for the note.

9. The only mention of code availability was the link to github ref 31 “Yoo-Jin Ha, J.K., Seungseok Kang, Junhan Kim, Se-Young Jo, and Sangwoo Kim Benchmark for improved accuracy in mosaic variant calling. GitHub (2021).” I could not find this github site when searching for it.

Answer to Q9 – We sincerely apologize for not providing a proper link for the code sharing site. We corrected it in the manuscript within the newly added “Data Availability” section (page 22).

10. In this approach, my understanding is that there are many more mosaic variants than there would be in a real sample, which would result in substantially higher estimates of precision than in a real sample. Instead of precision, maybe a better metric would be FPs/Mbp or even just # of FPs?

Answer to Q10 – We do agree with the reviewer and truly appreciate the brilliant suggestion. In this revision, we added the “FPs/Mbp” as a measure for absolute false positivity. We calculated FPs/Mbp for all analysis applicable and added in the main manuscript (line 106-108), Figure 2, and Supplementary Figures 2, 6 and 7. Briefly, we observed that MF and MT2 accompanied the

highest rates of FPs along with the highest sensitivities. Also, the advantage of the high-VAF (>25%) detection of MF and MH was extremely low FPs (please see the Figure below). We are happy to have this measure that further improved our study, and again thank the reviewer.

11. Were any of the callers used to form the benchmark also evaluated in this work?

Answer to Q11 – We thank the reviewer for the question regarding data contamination. We confirm that the callers used to form the ground truth dataset (i.e., germ line variants of six cell lines) were *not* used in the evaluation. We used the intersection of “Strelka2 germline calling mode” and DeepVariant for the ground truth generation. The only possibly related tool that was evaluated in the benchmark is the “Strelka2 somatic calling mode”. However, we think Strelka2 germline and somatic calling modes are intrinsically different because they are built on different probability models². In addition, and more fundamentally, we are confident that the ground truth set is a universal true variant set *per se*, because it has been strictly prepared by using the intersection, so the set itself is not biased towards any callers. To confirm this, we applied another germline caller, HaplotypeCaller ploidy 2, to the original cell line data, and found that

99.8% of the calls were reproduced. Overall, we would like to note that there is no data contamination issue in this study.

12. How conservative is the benchmark? For example, does it mostly exclude repetitive regions like segmental duplications or other regions prone to systematic errors like homopolymers and tandem repeats?

Answer to Q12 – We think that our benchmark is conservative to focus on performance in the high-confidence genomic regions. We also think that performance in the low confidence regions such as repetitive regions and low complexity regions can be of interest to some readers, because these regions are error-prone and where most variant callers are fraught with. While yet unstructured, we have data on these low-confidence regions too. However, we decide to include only the high-confidence regions in this benchmark for two following reasons. First, unlike germline and somatic variant calling, no benchmark has been done for mosaic variant calling, so the primary aim should be focused on the core performance of the algorithms. We believe that variant calling in the low confidence regions is more influenced by empirical or sometimes arbitrary filtration procedures, which lie outside the core models and remain secondary to the main purpose of this benchmark. Second, unlike germline and somatic variant calling, there are many more factors that should be assessed, such as single/paired sample calling, sample-specific/shared variants, VAF levels, and VAF imbalances within samples. Also, we aimed to provide future directions for method developers with additional strategies. We feel that the manuscript is already overwhelmed by a series of analyses, and the inclusion of the low-confidence regions may dilute the main conclusions. As the reviewer inquired and might have implied, we discussed the conservativeness of this study and mentioned the analysis on the low confidence regions as a further study in Discussion (line 323-327). We thank the reviewer for the questions.

13. It should be clear in the abstract that the authors are testing exome sequencing data.

Answer to Q13 – We thank for the reviewer’s comment and revised our abstract to clarify that the benchmark is based on exome (line 20).

14. It’s not clear to me how coverage variability in exome sequencing affects these results. Do the authors exclude variants below a certain coverage or deal with edges of captured regions in a different way?

Answer to Q14 – We appreciate the important question raised by the reviewer. We agree with the reviewer that capture bias exists in exome sequencing, and the actual sequencing data show read-depth variability within and across the exon regions. However, we did not specially exclude or stratify genomic regions by read coverage but considered the exome sequencing data as a whole entity to provide a real-world level interpretation. Thus, the overall performance is given for sequencing data of a certain target coverage. We believe that this is the most direct information for the readers who would like to refer to our study for their selection of mosaic variant strategies.

Nevertheless, to directly answer the reviewer's question about the effect of coverage variability, we newly added an analysis. We divided the ground truth data into three subsets based on base coverage: lowest 25%, medium 50%, and highest 25%, which corresponded to $<360\times$, $360\times$ to $803\times$, and $>803\times$, respectively. As expected, coverage unevenness was observed. We then compared the sensitivity, precision, and F1-score between the three subsets.

The results showed that the effect of coverage variability depends on the VAF (please see the below Figure). For the low VAF variants ($< 10\%$), the overall performance (F1-score, rightmost column) was higher in the high coverage regions. In contrast, we did not observe a significant effect in the medium and high VAF variants. Overall, it is true that the mosaic variant calling performance is variable within the exon regions in the same sequencing data, but the effect is limited to low-VAF variants. We added the analysis in Supplementary Notes (section 5) and discussed the effect of coverage variability in the Discussion (line 317-319). We again thank the reviewer for the question.

15. When using a benchmark to evaluate methods it is important to ensure it is reliably identifying FPs and FNs. Could the authors manually curate a subset of the FPs and FNs from each callset in a genome browser like IGV to ensure these are actual errors in the callset and not errors in the benchmark or how the authors ran the tools or performed comparisons?

Answer to Q15 – We appreciate this valuable comment. We do agree with the reviewer that this manual curation process is important to secure the robustness of the ground truth dataset, and we newly applied it in this revision.

First, we conducted the manual curation of FPs. The point of this curation is to check *if the FPs were actually present in the original cell line sequencing data*. If so, we can conclude that ground truth missed the variants as true positives and should be rescued.

There were 1,992 FPs from all tools. Due to the excessive number of manual curations, we applied a tiered approach, wherein FPs commonly called by callers have higher chances to be true variants. There were 10 FPs called by three callers (group 1), 424 by two callers (group 2), and 1,558 by one caller (group 3). We initially conducted all the group 1 FPs (10 FPs) and 10%

(45 FPs) of group 2 using IGV. The IGV alignment was assessed by three independent authors. Inspection of the 55 FPs confirmed that there is no sufficient evidence for the existence of the FP variants in the original cell line sequencing; the VAFs were too low to be germline variants (mostly up to three reads), and the base qualities were generally low for the alternative alleles. Based on the data, we concluded that all the FPs were truly FPs from the callers, and there were no errors in the ground truth set, without proceeding to the caller-specific FPs (group 3). We provide the screenshots of the 55 FP sites in a separate file for review.

We further investigated the source of the alternative alleles that caused FPs. We prepared an independent internal data set (three matched-normal exome sequencing data) and assessed the genomic regions of the reported FPs. We found that the independent data contained the same alternative alleles in 21 FP regions and were accompanied by frequent soft clipping and strand bias, which indicate that the FPs occurred in genomic positions of low quality, which are prone to alignment errors. Alternative alleles in the remaining 34 FPs were sample-specific and of extremely low VAF (mostly 1%), which we believe to be typical sequencing errors.

Next, we conducted the manual curation of FNs. The point of this curation is to check *if the FNs were actually absent in the original cell line sequencing data*. If so, we can conclude that ground truth falsely included the variants as true positives and should be removed.

For the confirmation of FNs, we first re-assessed the VAF of the positions in the original cell line data to check if there were sufficient allele frequencies to be germline variants. For all 18,873 control positive sites, we found that all the positions showed $VAF > 25\%$ (please see the Figure below). Together with the robustness of the germline variant calling process in the original cell line sequencing (please refer to the answer to Q11), we are convinced that all the FN calls were truly true variants in the benchmark dataset.

We also investigated the source of the FNs. We then found that sequencing data of the cell line mixtures may not contain the alternative alleles, even if the original cell line had, particularly when only a small portion of DNA was mixed to generate low VAF variants. For instance, we found that 2% and 17% of the true positive sites had no alternative alleles for 1% and 0.5% mixtures, in 1,100x coverage. It is true that no tools can call any true variants within these sites theoretically. Therefore, these FNs are not true FNs that come from variant calling performance and should be marked.

We thought of two scenarios to remark these FNs. Although undetectable, the final performance should include these FNs to represent the real-world level interpretation, because the same issue will be reproduced for all real analyses. Therefore, we calculated the theoretical upper limits of the calling performance for all evaluations of sensitivity and F1-score to separate the FNs

originated by miscalling from ones that are theoretically undetectable (please see the Figure below for example). We marked the upper limits in Figures 2b, 2c, 3e, Supplementary Figures 2, 6, and 7, and added related descriptions in the revised manuscript (Figure 2 legend and Supplementary Notes section 2). We appreciate the reviewer’s comments that improved the reliability of our analysis.

16. In the title, I think it would be more accurate to say “Extensive benchmarking (or evaluation)” rather than “Robust benchmark” because the focus of this manuscript is on the performance evaluation rather than making a benchmark, and it is not clear how robust the approach is given its use of germline variants.

Answer to Q16 – We appreciate the reviewer’s suggestion. We changed the title to “Comprehensive benchmarking and guidelines of mosaic variant calling strategies”. Although we clarified that the use of germline variants does not raise a substantial concern in Q5, we generally agree with the reviewer that the new title more clearly describes the main contents of the paper, which are the performance evaluation and practical guidelines for researchers and method developers. We thank the reviewer for the suggestion.

17. It appears that 3 of the 5 cell lines used in this work don’t have public exome or genome sequencing data apart from the authors’ work. One appears to only have restricted access for exome data, so others have determined that consent is not appropriate for public access (<https://www.ncbi.nlm.nih.gov/sra/?term=HBEC3-KT>). Informed consent is particularly

important if these are to be established as widely-used benchmarks. Could the authors clarify what human subjects review that was done to determine that these data could be made public, and whether the individuals from whom these cell lines were derived were consented for public release of genomic data?

Answer to Q17 – We appreciate the reviewer’s suggestion. We agree with the reviewer that informed consent is important to protect the confidentiality of genomic data, specifically if the genomic information is generated from specimens of identifiable donors. On the other hand, however, we used immortalized normal “cell lines”, and the information about the donor identity and the level of consent is mostly diminished. For example, one of the cell lines THLE-2 has been constructed in 1994, wherein the information about the source is only stated as “a collagenase/dispase perfusion of the left lobe from adult biopsy with no clinical evidence of cancer”. We believe that informed consent about the use of genomic information has not been even conceptualized at these times, and tracking the subject is no more available. Thus, reviewing the human subject and the consent to public release is a very difficult task or even impossible. We are attaching the informed consent policy of two cell lines that we could retrieve from the commercial cell line providers (ATCC) in separate files.

In this circumstance, we thought that the best standard of judgment could be only estimated by the general policy about handling the genomic information of cell lines, such as the ones manifested in the Cancer Cell Line Encyclopedia (CCLE) (“Consideration for Open Release of Genomic Data from Human Cancer Cell Lines”) that state the followings:

1. The risk to donors from publicizing genomic information is “exceedingly small” or “negligible” in commercially available cell lines.
2. Genetic data from commercially available cell lines are “already available openly” and “there is no additional risk.”
3. The benefit from releasing genomic information as “open-access” is substantially larger than the risk,

thereby concluding the open-access release of all genomic sequence and variation data “without requiring Data Use Certification”. We think that the same principle applies to the immortalized normal cell lines, and there will be minimal or no barriers to the use for the same purpose as ours. We are also confident that the genomic information from the mixed cell lines is even much safer from identification.

In addition, we think that the use of the genome sequencing data of the same or similar cell lines will not deteriorate even if they are available by controlled-access like other publicly available genomic data such as TCGA, ICGC, and a tremendous number of SRA (sequence read archive)

provided entries. We believe that the purpose of the controlled-access is not to restrict the use of the data from researchers, but to emphasize the ethical responsibility to the end-users.

Reviewer #3:

Remarks to the Author:

A. Summary of the key results

In this manuscript, Ha et al. performed a detailed benchmark of 11 variant calling strategies based on a benchmark dataset the same group generated previously (PMID 35115554) and provided their best practice and condition-dependent strength and weakness for users. One of the major conclusions, which is also commonly accepted in the field, is that different algorithm finds distinct parts of the true mosaic variants and can be complemented by others with different strategies. The authors also provided recommended best practices based on their benchmark dataset, which is very useful for the field, if their benchmark dataset is reflecting real-world data features.

B. Originality and significance: if not novel, please include reference

Calling postzygotic mosaic mutations (from non-cancer tissues) is very challenging, and is very important in this field. A comprehensive benchmark for different mosaic mutation calling strategies is very important not only for the mutation field, but also for researchers studying developmental biology, clinical genetics, and beyond. The originality of the study is valid as no benchmark for non-cancer mosaic mutations is carried out for different strategies other than tool builders themselves.

C. Data & methodology: validity of approach, quality of data, quality of presentation

While the data looks very informative based on their benchmark dataset, and calibrations are carefully carried out, I have some concerns:

Answer to the General comment:

We greatly appreciate the overall positive evaluation of our study and thank the reviewer for understanding the value of this benchmark. In this revision, we thoroughly addressed the concerns of the reviewer with additional analyses and clarification and reflected them in the revised manuscript. In particular, we applied the benchmark procedure on three additional independent biological datasets, confirming the overall consistency and the validity of the best practice guideline including suggested improvement strategies. We especially thank the reviewer for the suggestion. We hope that the revised manuscript is publishable and look forward to sharing our study with researchers in the relevant fields.

Q1. a. The benchmark dataset is generated by mixing 1100x WGS from cell lines with different haplotypes, will this affect the performance of any of the strategies?

Answer to Q1 – We appreciate the reviewer’s question. As the reviewer asked, we thoroughly considered all the possible biases that originated from the way we generated the benchmark dataset and may influence the evaluation. In conclusion, we are convinced that the benchmark results have not been affected by it, and most of the possible biases have been successfully treated, as additionally proven in the validation on three independent biological datasets (please see the answers to Q11).

The possible concerns that we considered were (1) modified presentation of germline variants by cell line mixing, (2) application to algorithms that use haplotype phasing, or (3) noise that originated from the source cell lines. (1), we resolved the modified germline composition issue by preparing a separate set of reference standards (Set B) that recovered the germline variants. W.r.t. (2), MosaicForecast has two models, haplotype phasing and using read features, among which the former model could be seriously affected, but was not evaluated in the benchmark. The MosaicForecast authors recommended their read feature model over the phasing model. W.r.t. (3), we thought that models that utilize information from the surrounding regions of variant candidates could be confounded by additional noise-like alternative reads that originated from the source cell lines. We conducted additional analysis in this revision and concluded that the effect of the noise is partial and small enough make our conclusions reliable (please see **Supplementary Notes** section 6). Overall, we believe that our benchmark is practically the most robust and sophisticated based on a large-scale ground truth set.

Q2. The features provided in Figure 4, as well as lines 209-213 also bring up a concern that haplotype-based features might bias this assessment. Do the authors have an independent biological dataset other than the simulated dataset to validate their conclusions?

Answer to Q2 – We understand the reviewer’s concern. We would like to clarify that the haplotype-based features we mentioned in the discussion are not the features that we utilized in the main evaluations, but the potential additional features that can further classify the false positives from the tools. Therefore, it did not cause any bias in the evaluation. In addition, we conducted additional benchmarks with three different independent biological datasets and confirmed the benchmark results to observe a consistent result (please refer to Q11).

Q3. Also, Supplementary Figure 1 showed distributions of VAFs for true positive and true negatives, are the distributions of these VAFs resembling VAFs from the real data? Are VAFs

the only features of the benchmark dataset that is similar to existing data, such as published human exomes sequenced in developing tissues or cancer?

Answer to Q3: We agree with the reviewer's concern about the representativeness of the VAF distribution for true positives and negatives that are used in this benchmark. We also seriously considered the composition in the preparation steps and tried to take the most realistic distribution that we could possibly conclude.

Nevertheless, as the reviewer might agree, determining the reference VAF distribution for the real mosaic variants is extremely difficult due to the lack of confirmed mosaic variants and distinct settings from individual studies. To check the possibility, we compared five independent biological datasets: Kim et al. 2022³, Breuss et al. 2020⁴, Wang et al. 2021⁵, Hsieh et al. 2020⁶, and Breuss et al. 2022⁷, which reported 130, 132, 400, 143, and 2256 mosaic variants, respectively. Drawing VAF distributions of those studies revealed remarkable inter-study variability (please see the figure below). For example, mosaic variants from Breuss et al. 2022 (study in brain) were mostly enriched in low VAF (<5%), whereas ones from Hsieh et al. 2020 (study in heart) were distributed in a wider VAF range (5-25%). The only common characteristic was the general enrichment in low VAFs and the rapid decrease in the number of high VAFs. We found that the VAF distribution of our benchmark dataset is located at the average of the five biological datasets. Therefore, while we cannot confirm the representativeness, we can conclude that our dataset sufficiently reflects the generally accepted VAF distribution of real mosaic variants. We thank the reviewer for the valuable comments and added the additional analysis in the **Supplementary Note** (section 8).

Q4. b. The author provided a thorough comparison of different strategies, do they have a reason for not including CaVEMan (PMID 34433962 and 34433962) which is used in a good amount of high-impact publications?

Answer to Q4: We thank the reviewer for the valuable suggestion. The exclusion of CaVEMan was based on our selection criteria: (1) algorithms that explicitly aim to detect mosaic mutations, (2) procedures previously used to discover mosaic variants, and (3) algorithms that can be adapted for mosaic mutation detection via simple usage modification. CaVEMan is basically a somatic variant caller that primarily targets medium-to-high VAF mutations from micro-dissected low-input low-depth WGS sequencing (a popular approach from Wellcome Sanger Institute); the paper that the reviewer referred to is such kind of study (please note that the reviewer cited same paper twice). In addition, CaVEMan to not target variants with VAF < 10% by its default setting (it classifies these variants as contamination), whereas our benchmark highlights low VAF variants (<5%). Therefore, we thought that CaVEMan does not satisfy the selection criteria.

Nevertheless, we newly tested CaVEMan for the benchmark data to confirm its performance of sample-specific variant calling. As expected, we observed very low sensitivity at <10% VAF (figure below). Also, the overall performance was low, probably because CaVEMan is optimized in low-depth sequencing data. We think the low performance of CaVEMan is caused by its unintended use, so we finally decided not to include CaVEMan in this study. We appreciate the

comments.

Q5. c. The author showed convincingly that higher sequencing depth is in general increasing the performance of mosaic variant detection. With the dropping of sequencing price, it will be more feasible to sequence all the samples at a much higher depth, most of the currently existing exome and genome data, however, are way below the minimum depth tested in this manuscript (125X). For exomes, most are below 100x, for clinical diagnostic exomes that are massively generated they are normally below 60x. Do the authors not recommend using any of the methods in the existing data or recommend to re-sequence the existing samples for mosaic detection? Do they have some estimates on existing low-depth samples?

Answer to Q5: We thank the reviewer for the insightful question. We do agree with the reviewer and newly tested on 60x down-sampled data in this revision. In conclusion, we recommend re-sequencing of the existing samples as the overall accuracy rapidly drops at this depth (please see the below Figures). From further inspection, we found that many 60x sequencing data do not contain alt-alleles that are crucial to call variants. A simple calculation shows that expected numbers of 1% and 5% VAF alt-allele at 60x data are 0.6 and 3, respectively; many will have less than these numbers or even zero (failed to be sequenced). This theoretically limits the accuracy for all tools (theoretical upper limits were denoted in dashed lines in the Figure). Consequently, we do not think that it is possible to use 60x for detecting mosaic variants in clinical diagnostic settings where robust precision is desired.

Q6. Also, for most of the parts in the manuscript, the authors didn't specify that they are working on ultra-deep exome data. Do the authors expect similar performance for genomes or capture panels?

Answer to Q6: We thank the reviewer for the comment and apologize for the unclear description. We now explicitly specified the use of ultra-deep exome sequencing in the revised manuscript including the abstract (line 20 and 317-319).

For the second question, we expect that the reported variant calling performance will be similar or slightly higher in the whole genome- or panel-sequencing if subjected to the same target depth. The most distinct characteristic of exome sequencing is the inter- and within-exon coverage variability that is caused by capture bias (please refer to the figure below). This means that a certain number of low-depth regions always exists in WES regardless of the target depth, which causes reduced accuracy for variant calling. We newly added an analysis to measure the effect of the coverage variability in exome sequencing on the performance, suggested by another reviewer. Briefly, we observed a significant drop in accuracy at the low 25% coverage regions

(<360x in 1100x-targeted WES), whereas the increase of accuracy is limited even in the top 25% coverage regions. In contrast, coverage is more uniform in WGS for all genome and in panel-seq for targeted intervals, which reduces the portion of low-coverage regions, compared to WES. Therefore, assuming the same target-depth, slight increase in performance is expected in genome or panel-seq. We thank the reviewer for the important question, and we revised our manuscript in Discussion (line 319-323) and Supplementary Notes (section 5).

Q7. Line 85: citation 8 used ploidy 50, does the author use ploidy 200 to compensate the 1100x WES to get to lower AF, or do they have evidence that p200 performs better than p50?

Answer to Q7: We chose to use ploidy 20 and ploidy 200 in our benchmark based on the recommendation from a BSMN (Brain Somatic Mosaicism Network) paper⁵, which sets the ploidy to the 20% of the overall sequencing coverage. Therefore, we chose ploidy 20 and 200 in our evaluation, as our benchmark data ranged from 125× to 1,100×.

In this revision, we also newly tested two different ploidy settings, ploidy 50 and ploidy 100 to inspect the effect of the ploidy value on the variant detection performance. We found a clear trade-off between sensitivity and specificity. The higher ploidy option increased sensitivity but lost precision, and vice versa in the lower ploidy options (please see the Figure below). But the overall classification power (F1-score) remained similar. Therefore, ploidy options can be selected lower or higher from the 20% of the overall coverage depending on the research purposes. We added Supplementary Notes about the ploidy effect (section 1) with the relevant

analysis result and revised our manuscript in line 93-95 and 411-412. We thank the reviewer for the comments.

Q8. e. Line 170-172, is this performance different because MF is running based on M2 results? If MF is running based on HC-p20 or HC-p200 will it improve the performance?

Answer to Q8: The selection of Mutect2 raw call sets as an input of the MF was based on the recommendation by the authors⁸ (“We used MuTect2 in its tumor-only mode for its high sensitivity, but other algorithms can be used”). To answer the reviewer’s question, we newly tested and compared the MF performance with two different inputs, raw call sets of MT2 and HC-p200 (figure below). We found two times more false positives in HC-p200 call sets in the low VAF (<10%) area. We speculate that MF has been optimized with the MT2 input from its initial model training (with MT2 call sets). We discussed this result in the Supplementary Notes (section 9). We thank the reviewer for the question.

Q9. f. Line 373: “PASS” recommended in DM tutorial.

Answer to Q9: We apologize for the unclear description. We only used “PASS” calls from DeepMosaic and clarified in the revised manuscript (line 417-418)

D. Appropriate use of statistics and treatment of uncertainties

Most of the statistics are well performed in the current manuscript, but there is some inconsistency between the currently used data and the published data from the author, claiming to be the same dataset.

Q10. a. The ground truth set is a unique resource and the basis for this study, however, in the original publication (PMID 35115554) where the author built their reference standard, there are 386,613 positives and 35,113,417 non-variants, but in the current manuscript there are 390,153 positives and 35,208,888 negative positives, in lines 61-63, the authors might want to explain what are the differences between the two datasets that are described?

Answer to Q10: We apologize for the unclear description of the numbers of true positives and negatives. We only used the variants within high-confidence genomics regions that are outside of repeats and homopolymers for this benchmark. This reduced the number of true positives to 345,552 SNVs and 8,706 INDELs (in total 354,258) from 374,565 SNVs and 12,048 INDELs (in total 386,613) in the original paper. Likewise, the number of true negatives was reduced to 35,111,725 non-variants and 18,151 germline variants (in total 35,129,876), from 35,113,147 non-variants and 19,936 germline variants (in total 35,133,083) in the original paper. We now

use only the actual numbers that were used in this study (high confidence) in the main text, and better clarified the procedures in the revised manuscript (line 71-73 and 399-403).

E. Conclusions: robustness, validity, reliability

Q11. a. The benchmark analyses and the recommendations in box 1 are highly appreciated, from a user perspective, it would be very important to know whether the same strategy applies to real-world data, published by the authors (PMID 36121845) for example.

Answer to Q11: We sincerely thank the reviewer for the valuable and important question. As the reviewer suggested, we confirmed our benchmark results and suggested best practices with previously published independent biological datasets, containing answer sets that were clearly validated with orthogonal experiments. Also, three biological datasets having different types of mosaic variants were chosen for validation to cover various scenarios in mosaic variant detection (e.g., extremely low VAF variants (<5%) found in brain tissues or shared variants in both balanced and imbalanced VAFs). Consequently, we could reproduce most of the conclusions and recommendations (8 out of 12 were applicable) with a total of 662 (274 true and 388 false) answer variants.

First, we obtained deep sequenced (~500X) multi-organ WES data (Kim et al. PLOS Genetics 2022, **BioData1**)³, which provided 130 validated positions with 117 of them validated as true, where a majority of them were shared by two or more organs. Also, we utilized 250X WGS data of sperm and blood pairs from eight individuals (Breuss et al. Nature Medicine 2020, **BioData2**)⁴, 57 true shared mosaic variants. Lastly, we acquired 250X WGS data from a single neurotypical brain tissue followed by robust and extensive validation experiments, conducted by the Brain Somatic Mosaicism Network (BSMN) (Wang et al. Genome Biology 2021, **BioData3**)⁵. From this data, we could obtain 43 true positives with 357 false positives derived from various error sources. Detailed information of the three studies and their answer sets are described in the table below.

Independent biological dataset utilized for evaluation													
	Original study	Data type	Tissue	Original Detection method	Validation method	# Validated mutation		# Low VAF (<10%)		# Medium VAF (10%-25%)		# High VAF (> 25%)	
BioData1	Kim et al. 2022 Plos Genetics	~500x WES	 ● Brain ● Heart ● Liver 	 ● Mutect2 ● RePlow ● NeuSomatic 	Deep-targeted amplicon or Sanger sequencing	130		89		27		14	
						True	False	True	False	True	False	True	False
						117	13	78	11	26	1	13	1
BioData2	Breuss et al. 2020	200x WGS	 ● Sperm ● Blood 	 ● Triodenovo with trio data 		132		102		25		5	

	Nature Medicine				Targeted amplicon sequencing	True	False	True	False	True	False	True	False
						114	18	84	18	25	0	5	0
						400		359		18		23	
						True	False	True	False	True	False	True	False
BioData3	Wang et al. 2021 Genome Biology	250X WGS	● Brain	6 different analytic methods including ● MosaicHunter ● Mutect2 ● Strelka2 ● HaplotypeCaller ploidy 2-10	PCR amplicon-based deep sequencing validation and multiplex PCR-based targeted single-end resequencing assay	43	357	37	322	4	14	2	21

Then, we confirmed the detection performance in single sample analysis, relevant to the four of our recommendations as below.

Recommendation #2: MosaicForecast and Mutect2 are generally recommended for calling mosaic SNVs in a single sample. These tools are particularly strong in calling low-VAF mutations (<10%).

Recommendation #3: MosaicHunter showed low sensitivity but extremely high precision. Together with other callers, these calls can be used for strict filtering or prioritization.

Recommendation #5: Call set concordance between callers and read depths is very low. Therefore, finding overlaps from different callers to achieve high confidence is not recommended. The use of multiple callers should be composed in a way of assigning one to the best-performing VAF area.

Recommendation #12: Provided with numerous features, scores and filters from multiple algorithms, a complementary ensemble approach can be an efficient start to improve performance without developing a new algorithm from scratch.

As suggested in **Recommendation #2**, MosaicForecast and Mutect2 tumor-only mode showed superb performance among the six single-sample detection methods (MH, MF, DM, MT2-to, HC-p20, and HC-p200) in all three datasets (please see figure on left below). Indeed, their performances at low VAF (<10%) were clearly the best datasets. Importantly, in BioData3 which was most informative owing to the sufficient number of negative controls (89%), the precision was shown to be greatly consistent with our benchmark result, in which MF and MH had the highest precision in overall VAFs while the somatic and germline callers (MT2-to, HC) showed a rapid decrease in precision at high VAF (>25%) area. We also confirmed that MH showed low sensitivity but extremely high precision (**Recommendation #3**). The call set concordance between callers was once again confirmed to be low and it is not appropriate to take an intersection of multiple callers to achieve a high confidence call set (**Recommendation #5**,

figure on right).

Also, we demonstrated three examples of the feature-level recombination within the tested algorithms, showing that informative features from one algorithm can bring a complementary effect to another (**Recommendation #12**). Again, with real-world level false positives in

BioData3, we observed that 27% of the false positives from MT2-to call set could be removed when filtered by a MF feature (alt soft clip > 0.05), with only 3% loss of the true positives (see table below). Also, 18% of the false positives from HC-p200 call set could be removed without losing any true call. BioData2 could not be applied here due to the absence of false positives and BioData1 was not informative as it is high-precision data, showing the true and false answer ratio 12.5. We expect that this kind of feature-level strategy would be more feasible at the variant calling steps, where various false positives threaten accuracy of the call sets.

Feature-level recombination with biological data							
Data	Ratio of #TP and #FP	Original Call	Feature used for filtration	Removed true positives (removed/total)	Removed false positives (removed/total)	% Loss of True Positives	% Removed False positives
BioData1 [#]	12.5	MT2-to [#]	MF alt soft clip > 0.05	1/100	0/8	1%	0%
		HC-p200	MT2-to MFRL alt < 150	6/100	0/8	6%	0%
		HC-p200	MF HetLH > 0.25	0/100	0/8	0%	0%
BioData3	1.5	MT2-to [#]	MF alt soft clip > 0.05	1/32	6/22	3%	27%
		HC-p200	MT2-to MFRL alt < 150	0/32	0/22	0%	0%
		HC-p200	MF HetLH > 0.25	0/32	4/22	0%	18%

[#]high precision call set (ratio of #TP and #FP = 12.5)

Next, we also could validate the provided conclusions and three recommendations for paired-sample analysis, for both sample-specific and shared variant detection.

Recommendation #6: For sample-specific calling, current somatic callers (Mutect2 and Strelka2) outperform in overall VAFs including low VAF (<5%)

Recommendation #7: For shared variant detection, although MosaicForecast and Mutect2 showed good overall accuracy, we want to highlight that the best-performing tool for each VAF range varied. We recommend also trying out the best tools for the expected VAF-pair area if one exists (see Fig. 3f).

Recommendation #8: In the present situation, we recommend utilizing somatic callers (Mutect2 or Strelka2) for sample-specific and MosaicForecast with the same strategy for shared somatic variant

Among the three biological datasets, we could attain sample-specific variants from BioData1, where the VAFs of the 25 true variants were distributed evenly from low (<10%) to high (>25%) (please see table below). We confirmed that MT2 and STK2 showed the best F1-score across all

VAF ranges including low VAF (<10%), as suggested in **Recommendation #6** (please see figure below), implying that those somatic methods utilizing joint genotyping supported by matched controls are currently the best strategy for sample-specific mosaic variant calling.

Number and variant allele frequencies of sample-specific variant answer set						
Data	#Sample-specific mutations					
BioData1	Total					
	28					
	True			False		
	25			3		
	# Low VAF (<10%)	# Medium VAF (10%-25%)	# High VAF (> 25%)	# Low VAF (<10%)	# Medium VAF (10%-25%)	# High VAF (> 25%)
	7	9	9	2	0	1

Then, to evaluate shared variant detection performance, BioData1 and BioData2 could be utilized, supported by 56 and 28 answer variants respectively. The answer sets were comprised of true variants that were shared by two to six samples from an individual (see table below). Then, we again generated VAF-VAF combinations of shared variants, where variants in each

Number of shared variants in biological data										
Data	# Shared mutation									
BioData1	Total									
	56									
	True					False				
	51					5				
	# Shared tissue	2	3	4	6	# Shared tissue	2	3		
	# Mutation	18	2	2	1	# Mutation	1	1		
BioData2	Total									
	28									
	True					False				
	28					0				
	# Shared tissue	2				# Shared tissue	0			
	# Mutation	14				# Mutation	0			

sample could be partitioned based on four different VAF ranges: very-low (VL): $\leq 5\%$, low (L): $> 5\%$ and $\leq 10\%$, medium (M): $> 10\%$ and $\leq 25\%$, and high (H): $> 25\%$. After Partitioning the VAF ranges of a sample pair into 16 ($= 4 \times 4$) areas and assigning the sample with higher VAF to the x-axis, we could obtain 10 blocks of VAF combinations. The true and false shared variants could be located within four (BioData1) and six (BioData2) blocks (out of ten) and utilized for evaluation (please see table below).

As a result, we confirmed that only MF and MT2 maintained good overall accuracy (F1-score) in both datasets (please see figure below). Notably, several inevitable limitations existed while we validate the benchmark results with biological data, such as (1) the answer sets itself is already biased because it was often selected with one of the evaluated methods and (2) the answer set is highly precision-focused data which was conservatively chosen for validation experiments. For example, we observed MF showing both minimum (0) and maximum (1) F1-score at H-M (high-

medium) VAF block in each dataset, owing to the lack of answer where only one true variant could be assigned in both datasets, without a false positive. Also, MH, M2S2MH (MH with rescue strategy), and DM were shown to have remarkably high performance in only one of the data (BioData2). We assume that this is because this data was used for the training of DM model construction and possibly MH could have affected the original call sets in some way. Nevertheless, good performance MF and MT2 were reproducible in both datasets, as suggested in **Recommendation #7**.

Moreover, we also suggested trying out the best tools for the expected VAF-pair area in shared variant detection, as the best-performing tool in each VAF combination area highly varies (**Recommendation #7**). Herein, in both biological datasets, we were able to reproduce the original combinations of the best performing-ensemble approach, with consideration of the three limitations in the biological datasets: (1) Mutect2 was used for the call set construction for the BioData1, (2) lack of true positives in some combination blocks (VL-VL and H-M), and (3) overestimated performance of MH and DM in BioData2 (please see figure below).

In addition, we recommended using MF with a rescue strategy for shared variant detection (**Recommendation #8**). As expected, we confirmed that MF with rescue strategy achieved a remarkable performance as a single method with BioData2, covering all the VAF areas including the most challenging VL-VL (very low in both, <5%) area (please see figure below). BioData1 could not be applied because no misclassified variants existed.

Although most of the recommendations were able to be tested, **Recommendation #4** (*MosaicForecast is considered the current best algorithm for mosaic INDEL calling, whereas no algorithms are successful in calling low-*VAF* (<4%) INDELS*) couldn't be assessed here, as the answer sets in biological data were limited to mosaic SNVs. Also, BioData1 was the only multi-sample (≥ 3) data to test the utilization of the developmental lineage tree (**Recommendation #9**), it was highly limited to demonstrate as it only contained two shared false positives in the answer set.

In summary, we clearly reproduced our main conclusions and all the applicable recommendations, with three independent biological datasets containing various forms of mosaic variants. In this way, we proved that our benchmark results and practical suggestions are highly applicable to real-world data, covering diverse forms of mosaic variants and their accurate detection. We newly added relevant analyses methods and results in **Supplementary Notes**, directed by discussion in manuscript (line 276-268). We greatly appreciate the reviewer's comments that improved the reliability of our analysis.

F. Suggested improvements: experiments, data for possible revision

Q12. a. Comparison with CaVEMan as an important independent strategy

Answer to Q12: Already described in **Q4** of the reviewer 3.

Q13. b. Inclusion of experimental data (PMID 36121845 for example) to validate the major conclusions or additional data to support that the benchmark dataset is providing enough features that resemble real data.

Answer to Q13: Already described in **Q11** of the reviewer 3.

G. References: appropriate credit to previous work?

Q14. a. This study focuses on SNV/INDELS, the author might want to point this out so that readers can get better use of their benchmark, and also maybe specify do these strategies also work on non-human organisms (PMID: 35418684).

Answer to Q14: We appreciate the reviewer's comment and revised abstract and introduction in line (21 and 48) to point out that the benchmark is focusing on mosaic SNV/INDEL detection. Also, we specified that our benchmark results can broaden our scientific understandings of human and non-human model organisms in the revised manuscript (line 346-348).

H. Clarity and context: lucidity of abstract/summary, appropriateness of abstract, introduction and conclusions

Overall the clarity is good, I have some minor comments:

Answer to Q15: a. Line 17: I think mosaic variants can exist in germline and somatic tissues, so this sentence is not very accurate. Heterozygous, homozygous, and mosaic variants are more comparable concepts?

Answer to Q15: We agree with the reviewer that the form of genomic variants can be divided as hetero-, homozygous, or mosaic variants. Thanks to the reviewer's suggestion, we revised our manuscript in line 17.

Answer to Q16: b. Line 18: The focus of different studies is affecting the use of methods, not sure chaotic is a good word for the field

Answer to Q16: Thanks to the reviewer's comment, we changed the word "chaotic" to "disorganized" in the manuscript line 18.

Answer to Q17: c. Figure 3: What's the depth shown here? Also in the legend: log10 scale

Answer to Q17: The results shown in Figure 3 were based on the 1,100× data and we revised that the data shown in the Figure 3 is based on 1,100× depths in the legend. We also corrected the typo log10 scale.

References

1. Bohrson, C.L. *et al.* Linked-read analysis identifies mutations in single-cell DNA-sequencing data. *Nat Genet*

- 51, 749-754 (2019).
2. Kim, S. *et al.* Strelka2: fast and accurate calling of germline and somatic variants. *Nat Methods* **15**, 591-594 (2018).
 3. Kim, J.H. *et al.* Analysis of low-level somatic mosaicism reveals stage and tissue-specific mutational features in human development. *PLoS Genet* **18**, e1010404 (2022).
 4. Breuss, M.W. *et al.* Autism risk in offspring can be assessed through quantification of male sperm mosaicism. *Nat Med* **26**, 143-150 (2020).
 5. Wang, Y. *et al.* Comprehensive identification of somatic nucleotide variants in human brain tissue. *Genome Biol* **22**, 92 (2021).
 6. Hsieh, A. *et al.* EM-mosaic detects mosaic point mutations that contribute to congenital heart disease. *Genome Med* **12**, 42 (2020).
 7. Breuss, M.W. *et al.* Somatic mosaicism reveals clonal distributions of neocortical development. *Nature* **604**, 689-696 (2022).
 8. Dou, Y. *et al.* Accurate detection of mosaic variants in sequencing data without matched controls. *Nat Biotechnol* **38**, 314-319 (2020).

Decision Letter, first revision:

Our ref: NMETH-AS51120A

16th Jun 2023

Dear Dr. Kim,

Thank you for submitting your revised manuscript "Comprehensive benchmarking and guidelines of mosaic variant calling strategies" (NMETH-AS51120A). It has now been seen by the original referees and their comments are below. The reviewers find that the paper has improved in revision, and therefore we'll be happy in principle to publish it in Nature Methods, pending minor revisions to satisfy the referees' final requests and to comply with our editorial and formatting guidelines.

TRANSPARENT PEER REVIEW

Please note: we allow redactions to authors' rebuttal and reviewer comments in the interest of

confidentiality. If you are concerned about the release of confidential data, please let us know specifically what information you would like to have removed. Please note that we cannot incorporate redactions for any other reasons. Reviewer names will be published in the peer review files if the reviewer signed the comments to authors, or if reviewers explicitly agree to release their name. For more information, please refer to our [FAQ page](https://www.nature.com/documents/nr-transparent-peer-review.pdf).

ORCID

Sincerely,

Lin Tang, PhD
Senior Editor
Nature Methods

Reviewer #1 (Remarks to the Author):

The authors have addressed all my comments. Well done.
I also think that they have provided a comprehensive response to the critique and questions by other reviewers.

Reviewer #2 (Remarks to the Author):

The authors have done a good and comprehensive job responding to the reviewers' comments. My only remaining comment is to strongly encourage the authors to discuss with an expert in human subjects about whether their data should be in controlled-access or public data repositories. The NCI CCLE guidance is quite old and this area is quickly evolving, including new guidance from the NIH about genomic data sharing at <https://sharing.nih.gov/genomic-data-sharing-policy>.

Reviewer #3 (Remarks to the Author):

In the revised manuscript, the authors addressed major concerns with additional analyses of biological datasets, as well as additional performance analyses. The manuscript and its figures are significantly improved.

Minor:

line 122, mention of Extended Data Figure 3c before 3a and 3b.

Author Rebuttal, first revision:

Reviewers' Comments:

Reviewer #1:

Remarks to the Author:

The authors have addressed all my comments. Well done.

I also think that they have provided a comprehensive response to the critique and questions by other reviewers.

Answer for Reviewer #1:

We sincerely appreciate all the reviewer's comments and questions that highly improved our manuscript.

Reviewer #2:

Remarks to the Author:

The authors have done a good and comprehensive job responding to the reviewers' comments. My only remaining comment is to strongly encourage the authors to discuss with an expert in human subjects about whether their data should be in controlled-access or public data repositories. The NCI CCLE guidance is quite old and this area is quickly evolving, including new guidance from the NIH about genomic data sharing at <https://sharing.nih.gov/genomic-data-sharing-policy>.

Answer for Reviewer #2:

We greatly appreciate all the questions and comments from this reviewer. About the remaining concern, we discussed with a few experts including prof. Yu Rang Park (Yonsei University College of Medicine), who is the leader of ISO/TC 215/SC 1/AHG1 (Secondary use of genomic data) and developed the ISO/PRF TS 23357 Genomics Informatics (Clinical genomics data sharing specification for next-generation sequencing). Overall, these experts were in complete

agreement that there is no concern about publicly sharing genomic sequencing data of cell lines because (1) the cell line sequencing data are considered already publicly available, and (2) personal identification of the cell line donor is considered unavailable, or providers already acquired agreements from the donors in case identification is concerned. The only exception was HBEC30-KT, which is under a commercial license from the NIH and UT Southwestern. We checked that the original cell line sequencing data from the author (prof. John Minna) is under controlled access. Therefore, we changed the access-type of our HBEC30-KT sequencing from public to controlled-access (it may take days to be updated in SRA).

Reviewer #3:

Remarks to the Author:

In the revised manuscript, the authors addressed major concerns with additional analyses of biological datasets, as well as additional performance analyses. The manuscript and its figures are significantly improved.

Minor:

line 122, mention of Extended Data Figure 3c before 3a and 3b.

Answer for Reviewer #3:

We are grateful to the reviewer's suggestions and comments. As commented, we switched the panels within Supplementary Figure 3 to mention them in a right order.

Final Decision Letter:

12th Sep 2023

Dear Professor Kim,

I am pleased to inform you that your Analysis, "Comprehensive benchmarking and guidelines of mosaic variant calling strategies", has now been accepted for publication in Nature Methods. Your paper is tentatively scheduled for publication in our December print issue, and will be published online prior to that. The received and accepted dates will be 11th Dec 2022 and 12th Sep 2023. This note is

intended to let you know what to expect from us over the next month or so, and to let you know where to address any further questions.

Over the next few weeks, your paper will be copyedited to ensure that it conforms to Nature Methods style. Once your paper is typeset, you will receive an email with a link to choose the appropriate publishing options for your paper and our Author Services team will be in touch regarding any additional information that may be required.

Please note that *Nature Methods* is a Transformative Journal (TJ). Authors may publish their research with us through the traditional subscription access route or make their paper immediately open access through payment of an article-processing charge (APC). Authors will not be required to make a final decision about access to their article until it has been accepted. [Find out more about Transformative Journals](https://www.springernature.com/gp/open-research/transformative-journals)

Authors may need to take specific actions to achieve [compliance with funder and institutional open access mandates](https://www.springernature.com/gp/open-research/funding/policy-compliance-faqs). If your research is supported by a funder that requires immediate open access (e.g. according to [Plan S principles](https://www.springernature.com/gp/open-research/plan-s-compliance)) then you should select the gold OA route, and we will direct you to the compliant route where possible. For authors selecting the subscription publication route, the journal's standard licensing terms will need to be accepted, including [self-archiving policies](https://www.springernature.com/gp/open-research/policies/journal-policies). Those licensing terms will supersede any other terms that the author or any third party may assert apply to any version of the manuscript.

Once your manuscript is typeset and you have completed the appropriate grant of rights, you will receive a link to your electronic proof via email with a request to make any corrections within 48 hours. If, when you receive your proof, you cannot meet this deadline, please inform us at rjsproduction@springernature.com immediately.

Your paper will now be copyedited to ensure that it conforms to Nature Methods style. Once proofs are generated, they will be sent to you electronically and you will be asked to send a corrected version within 24 hours. It is extremely important that you let us know now whether you will be difficult to contact over the next month. If this is the case, we ask that you send us the contact information (email, phone and fax) of someone who will be able to check the proofs and deal with any last-minute problems.

Once your paper has been scheduled for online publication, the Nature press office will be in touch to confirm the details.

Content is published online weekly on Mondays and Thursdays, and the embargo is set at 16:00 London time (GMT)/11:00 am US Eastern time (EST) on the day of publication. If you need to know the exact publication date or when the news embargo will be lifted, please contact our press office after you have submitted your proof corrections. Now is the time to inform your Public Relations or Press Office about your paper, as they might be interested in promoting its publication. This will allow them time to prepare an accurate and satisfactory press release. Include your manuscript tracking number NMETH-AS51120B and the name of the journal, which they will need when they contact our office.

About one week before your paper is published online, we shall be distributing a press release to news organizations worldwide, which may include details of your work. We are happy for your institution or funding agency to prepare its own press release, but it must mention the embargo date and Nature Methods. Our Press Office will contact you closer to the time of publication, but if you or your Press Office have any inquiries in the meantime, please contact press@nature.com.

Nature Portfolio journals [encourage authors to share their step-by-step experimental protocols](https://www.nature.com/nature-research/editorial-policies/reporting-standards#protocols) on a protocol sharing platform of their choice. Nature Portfolio's Protocol Exchange is a free-to-use and open resource for protocols; protocols deposited in Protocol Exchange are citable and can be linked from the published article. More details can found at www.nature.com/protocolexchange/about.

Please feel free to contact me if you have questions about any of these points. Thank you very much again for publishing your paper at Nature Methods!

Best regards,

Lin Tang, PhD
Senior Editor
Nature Methods

** Visit the Springer Nature Editorial and Publishing website at http://editorial-jobs.springernature.com?utm_source=ejp_NMeth_email&utm_medium=ejp_NMeth_email&utm_campaign=ejp_Nmeth for more information about our career opportunities. If you have any questions please click [here](mailto:editorial.publishing.jobs@springernature.com).**